# Molecular architecture of a cylindrical self-assembly at human centrosomes

Tae-Sung Kim [1], Liang Zhang [1], Jong Il Ahn [1], Lingjun Meng[1], Yang Chen[1], Eunhye Lee[1],
Jeong Kyu Bang [2], Jung Mi Lim[1], Rodolfo Ghirlando[3], Lixin Fan[4], Yun-Xing Wang[5], Bo Yeon Kim [6],
Jung-Eun Park [1] & Kyung S. Lee [1]

The cell is constructed by higher-order structures and organelles through complex interactions among distinct structural constituents. The centrosome is a membraneless organelle composed of two microtubule-derived structures called centrioles and an amorphous mass of pericentriolar material. Super-resolution microscopic analyses in various organisms revealed that diverse pericentriolar material proteins are concentrically localized around a centriole in a highly organized manner. However, the molecular nature underlying these organizations remains unknown. Here we show that two human pericentriolar material scaffolds, Cep63 and Cep152, cooperatively generate a heterotetrameric α-helical bundle that functions in conjunction with its neighboring hydrophobic motifs to self-assemble into a higher-order cylindrical architecture capable of recruiting downstream components, including Plk4, a key regulator for centriole duplication. Mutations disrupting the self-assembly abrogate Plk4-mediated centriole duplication. Because pericentriolar material organization is evolutionarily conserved, this work may offer a paradigm for investigating the assembly and function of centrosomal scaffolds in various organisms.

[1] Laboratory of Metabolism, National Cancer Institute, National Institutes of Health, Bethesda, MD 20892, USA. [2] Division of Magnetic Resonance, Korea Basic Science Institute, Ochang, Republic of Korea. [3] Laboratory of Molecular Biology, National Institute of Diabetes and Digestive and Kidney Diseases, National Institutes of Health, Bethesda, MD 20892, USA. [4] Basic Science Program, Frederick National Laboratory for Cancer Research, Frederick, MD 21702, USA. [5] Structural Biology Laboratory, National Cancer Institute, National Institutes of Health, Frederick, MD 21702, USA. [6] Chemical Biology Research Center, Korea Research Institute of Bioscience and Biotechnology, Ochang, Republic of Korea. These authors contributed equally: Tae-Sung Kim, Liang Zhang, Jong Il Ahn. Correspondence and requests for materials should be addressed to K.S.L. (email: kyunglee@mail.nih.gov)

The architecture of a living organism is established through varying degrees of hierarchical self-organization, from single molecules to macromolecular assemblies. Investigating how proteins interact with each other to form a higher-order structural entity that can offer a new layer of biological functions is a key step to unlocking the mystery of life.

The centrosome, which plays a central role as the main microtubule (MT)-organizing center for animal cell division, is a membraneless organelle composed of a pair of orthogonally arranged MT-derived apparatus, called centrioles, and the surrounding pericentriolar material (PCM)[1,2]. In various eukaryotic organisms, PCM proteins are concentrically arranged around a centriole in a highly organized manner[3–6]. Based on this observation, proper positioning and organization of PCM proteins may be important for promoting different cellular processes in a spatially regulated manner. Not surprisingly, aberrations in the function of PCM scaffolds are associated with many human diseases, including cancer, microcephaly, ciliopathy, and dwarfism[7,8].

At present, how this membraneless cellular architecture is assembled and how the centrosomal scaffold proteins display a cylindrical pattern of localization around a centriole remain unknown. The formation of a higher-order structure requires biochemical and biophysical properties that allow a building-block molecule to establish the boundary of an assembled structure in an open cytosolic environment. Recent studies showed that a *Caenorhabditis elegans* centrosomal scaffold, Spd5, homomerizes to form a network-like assembly through its phase-separating ability[9,10]. In addition, the *D. melanogaster* Centrosomin (Cnn) appears to form a network-like assembly that does not undergo dynamic turnover[11]. However, both Spd5 and Cnn do not appear to form an organized higher-order architecture with a distinguishable morphology.

Here, we demonstrate that two of the human PCM scaffolds, Cep63 and Cep152[12–14], cooperate to generate a higher-order cylindrical self-assembly capable of recruiting their downstream components, such as Plk4, a key regulator of centriole duplication[15–18]. Cooperative self-assembly of a three-dimensional cylindrical architecture by two distinct intracellular proteins is striking. Since Cep152 is thought to function as a licensing factor for Plk4-mediated centriole biogenesis[19], investigating the mechanism of how Cep152 functions in conjunction with Cep63 to self-assemble a macroscale structural entity will be important to comprehend the centrosomal architecture critically required for Plk4-dependent centriole duplication.

## Results

### Coiled coil-mediated cooperative Cep63–Cep152 interaction.

Cep152 is composed of several coiled-coil domains covering a large portion of the protein (Fig. 1a) that appear to be stretched out over 100 nm in length[6]. Cep63, which has been reported to interact with Cep152[14,20], also contains multiple predicted coiled coils (COILS Server) (Fig. 1a). Given that coiled-coil structure is a highly versatile protein-binding motif that can assemble into diverse subcellular structures[21], Cep63 and Cep152 may play an important role in the organization of a higher-order centrosomal architecture.

Coimmunoprecipitation analyses with transfected lysates revealed that the two C-terminal coiled-coil motifs of Cep63 (residues 490–541; named P1b) and Cep152 (residues 1205–1272; named M4e) were necessary and sufficient for their interactions in vivo (Fig. 1a and Supplementary Fig. 1a, b). This finding concurs with previous studies[22,23]. Cep63 P1b and Cep152 M4e are predicted to form a single α-helical fragment with enriched hydrophobic residues (i.e., hydrophobic motifs) at opposite ends

of the two fragments (Supplementary Fig. 1c, d). Notably, although the major interaction affinity appeared to stem from the Cep63 P1b5(502–541) and M4e13(1205–1260) regions, the loss of their hydrophobic regions significantly (~40%) impaired the P1b–M4e interaction (Supplementary Fig. 1e, f).

Additional studies showed that the N-terminal (1–220) region and, weakly, the C-terminal (400–541) region both exhibited the capacity to mediate Cep63 homomerization (Supplementary Fig. 1g). Similarly, the N-terminal (1–512) region and, weakly, the middle M4(790–1380) region both possessed Cep152 homomerization activity (Supplementary Fig. 1h). While Cep63 full-length (FL) interacted with Cep152 FL efficiently (Supplementary Fig. 1i), the provision of Cep63 or Cep152 greatly (~8-fold) enhanced the homomeric Cep152–Cep152 or Cep63–Cep63 interaction, respectively (Fig. 1b and Supplementary Fig. 1j). A similar level of cooperativity was observed in the shorter Cep63 P1b(490–541)–Cep152 M4e(1205–1272) interaction (Fig. 1c and Supplementary Fig. 1k). Thus, the P1b and M4e fragments possess the full capacity to recapitulate the heteromeric Cep63–Cep152 interaction (Fig. 1d). Consistently, Cep63 P1b and Cep152 M4e were necessary and sufficient to localize to centrosomes in vivo (Fig. 1e and Supplementary Fig. 1l). Cep192, which is reported to bind to Cep152[24], exhibited only a low level of interaction when compared with Cep63 (Supplementary Fig. 1m), thus diminishing the likelihood of significantly altering the cooperative formation of the Cep63•Cep152 complex.

### Hydrophobic motif-dependent clustering of Cep63 and Cep152.

To investigate the biochemical nature of the Cep63–Cep152 interaction, we purified the Cep63 P1(424–541) •Cep152 M4d(1205–1295) complex from *Escherichia coli* (the M4e(1205–1272)-containing complex was highly insoluble) and performed sedimentation velocity analysis (Fig. 2a). At 10 μM, the Cep63 P1•Cep152 M4d complex sedimented as a single species at 2.83S with an estimated molecular weight (MW) of 51 kDa, suggestive of a 2:2 heterotetrameric P1•M4d complex (expected MW of 51.92 kDa). Interestingly, as the concentration was raised, the sedimentation coefficient for this species increased slightly, and at 90 μM, a distinct faster-sedimenting species was observed. The signal contribution and sedimentation coefficient of the faster-sedimenting species (~4S) increased at 245 μM, indicating the presence of a higher-order complex that was in fast exchange with the 2:2 complex. This could represent a 4:4 Cep63 P1(424–541)•Cep152 M4d (1205–1295), although the exact nature of this species remains to be identified. This finding suggests that the complex may self-associate and generate a higher-order complex in a concentration-dependent manner. Hydrophobic residues are known to drive protein–protein interactions because of their propensity to cluster in an aqueous environment[25,26]. Consistent with this notion, the Cep63 P1(4A)•Cep152 M4d(5A) complex, bearing Ala mutations in their hydrophobic motifs (L497A, I500A, F504A, and L505A in P1 and L1260A, I1261A, L1263A, I1266A, and L1267A in M4d), remained only as a single species of ~2.74S with an estimated MW of 49 kDa (expected MW of 50.92 kDa for the 2:2 P1(4A)•M4d(5A) complex) even at concentrations as high as 500 μM (Fig. 2a). This is clearly noted in a plot of the weighted-average sedimentation coefficients $S_w$, where the P1•M4d complex, but not its hydrophobic mutant, exhibits some form of higher-order self-association (Fig. 2b). The 4A and 5A mutations did not appear to disrupt the secondary structure of the P1•M4d complex (Supplementary Fig. 2a). Thus, the hydrophobic motifs present in the P1•M4d complex may play an important role in driving the complex to form a higher MW complex.

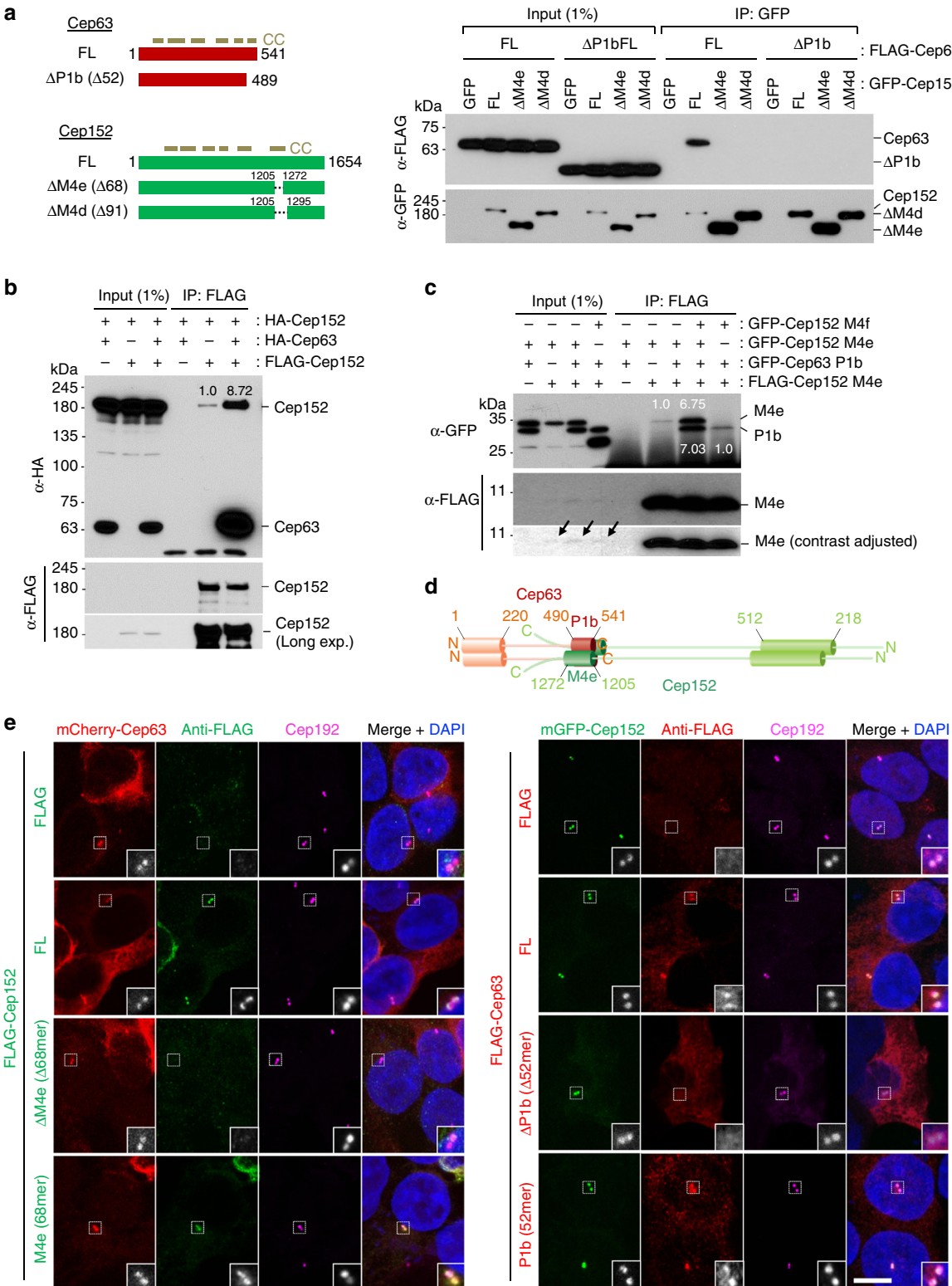

**Fig. 1** Mutual dependency of Cep63 and Cep152 for interaction and subcellular localization. **a–c** Immunoprecipitation (IP) and immunoblotting analyses using transfected HEK293 cells. FL, full-length; ΔP1b, Cep63 lacking residues 490–541; ΔM4e and ΔM4d, Cep152 lacking residues 1257–1272 and 1257–1295, respectively. Note that IP for **b** was carried out using a mixture of lysates from cells transfected separately with each of the indicated constructs and that the provision of HA-Cep63 greatly increased the level of coprecipitated HA-Cep152 (see also the Supplementary Fig. 1i legend). GFP-Cep152 M4f, which fails to bind to Cep63 (Supplementary Fig. 1b), serves as a control in **c**. CC, coiled-coil domains predicted by COILS server[58]; numbers, relative signal intensities. Arrows indicate input signals. **d** Schematic diagram showing the regions for the heterotetrameric Cep63–Cep152 and homomeric Cep63–Cep63 or Cep152–Cep152 interactions. **e** Confocal images showing immunostained transfected HEK293 cells. Dotted boxes, areas of enlargement. Bar, 10 μm. Quantified data obtained from three independent experiments are provided in Supplementary Fig. 1l

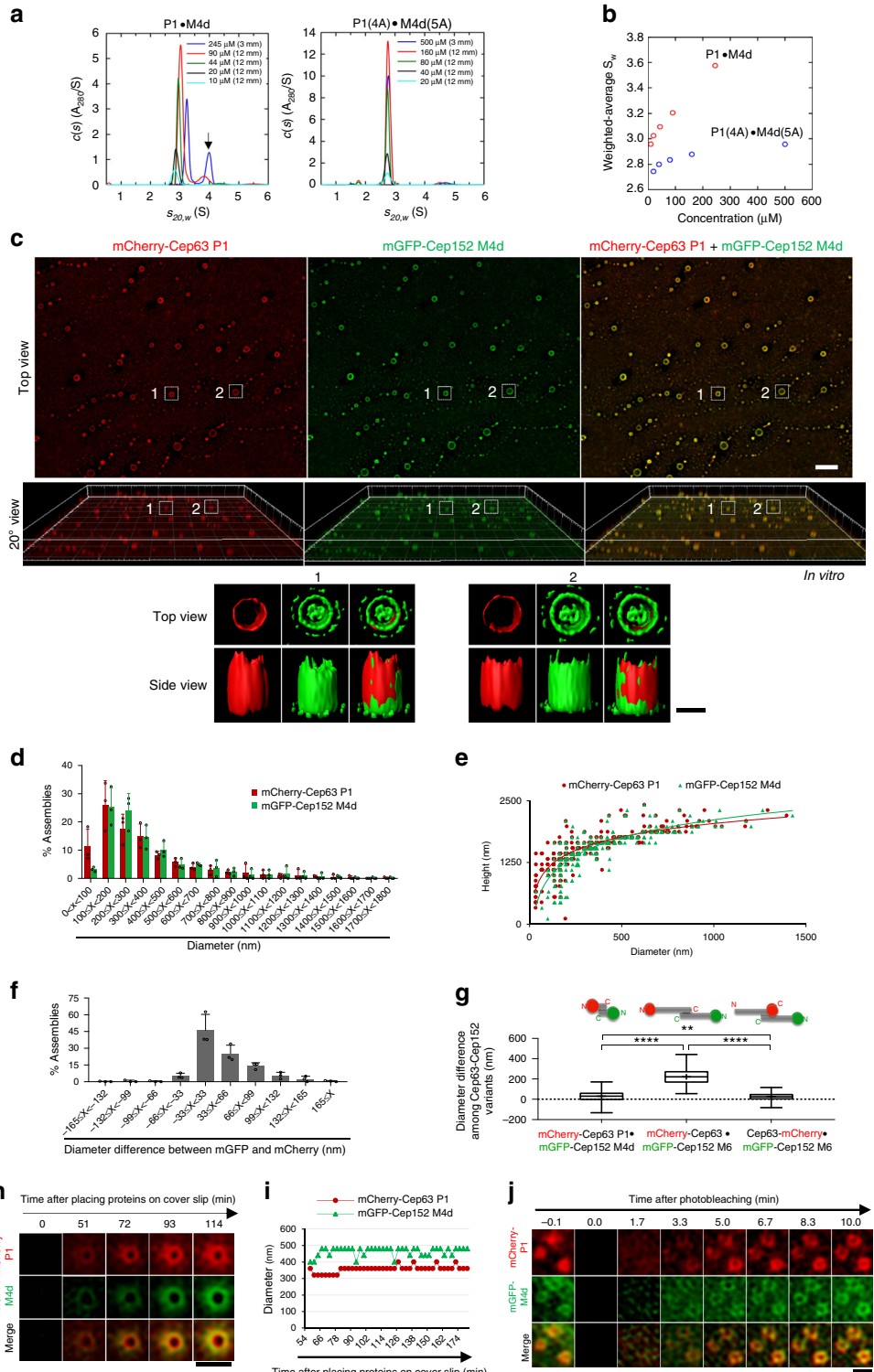

## A cylindrical self-assembly by the Cep63–Cep152 complex.

Cep152 displays an open-ended cylinder-like localization around a centriole[27]. It remains unknown whether Cep152 (or the Cep63•Cep152 complex) is recruited to a protein (or proteins) pre-arranged in a cylindrical shape around a centriole or if it has the ability to generate a cylindrical self-assembly. To investigate these possibilities, we purified a minimal Cep63•Cep152 complex (i.e., a monomeric mCherry-fused Cep63 P1 and a monomeric green fluorescent protein (mGFP containing the A206K mutation)[28] -fused Cep152 M4d) and its corresponding P1(4A)•M4d

(5A) hydrophobic mutant for comparative analysis. Three-dimensional structured-illumination microscopy (3D-SIM) revealed that the mCherry-Cep63 P1•mGFP-Cep152 M4d complex (6 μM) incubated at room temperature (RT) in a physiologically relevant buffer (20 mM Tris-HCl (pH 8.0), 150 mM NaCl) failed to significantly generate a microscopically detectable assembly. Since a centriolar structure may provide a spatial cue for Cep63 and Cep152 arrangement in vivo and the self-assembly process may occur only at concentrations greater than the self-assembling threshold, we incubated the same protein complex on

**Fig. 2** Cooperative formation of a dynamic cylindrical self-assembly by Cep63 and Cep152. **a**, **b** Sedimentation velocity analyses with Cep63 P1•Cep152 M4d or its respective P1(4A)•M4d(5A) mutant complex, using the complexes shown in Supplementary Fig. 2a, left. A high-molecular weight (MW) species (arrow) with rapid sedimentation velocity (~4S) detected at higher concentrations may likely represent a dimer of the ~2.74S heterotetramer (i.e., heterooctamer). Weighted-average sedimentation coefficients obtained from the $c(s)$ profiles are presented in **b** for the Cep63 P1•Cep152 M4d (red) and P1(4A)•M4d(5A) mutant (blue) complexes as a function of concentration. **c** Three-dimensional structured-illumination microscopy (3D-SIM (top) and surface rendering (bottom) of the in vitro self-assembly generated by the mCherry-Cep63 P1•mGFP-Cep152 M4d complex. Bar, 2 μm; bar for rendered image, 0.5 μm. **d**–**f** The dimension of the self-assemblies (**d**, **e**) and the diameter difference for mCherry-Cep63 P1 and mGFP-Cep152 M4d fluorescence (**f**) were determined from a total of 429 cylindrical assemblies obtained from three independent experiments. The Zeiss Zen software allowed to determine the inter-signal distance of up to 33nm/pixel (see Methods for details). A sharp drop in the number of small assemblies (<100 nm in diameter) in **d** is due to the resolution limit of SIM. The interpolated line of best fit is shown in **e**. Bars, s.d. **g** Summary of diameter differences between mGFP and mCherry fluorescence for the indicated cylindrical assemblies. Center line, median; cross, mean; box limits, upper and lower quartiles; whiskers, maximum or minimum of the data. **P < 0.01; ****P < 0.0001 (unpaired two-tailed $t$ test). **h**, **i** SIM-total internal reflection fluorescence (TIRF) time lapse of the self-assembling process of the mCherry-Cep63 P1•mGFP-Cep152 M4d complex in vitro (**h**) and the diameters of the assemblies as a function of time (**i**). Data are representative of a total of eight independent time-lapse movies analyzed. Bar, 1 μm. **j** Fluorescent recovery after photobleaching (FRAP) analysis for the mCherry-Cep63 P1•mGFP-Cep152 M4d self-assemblies in vitro. Representative data from a total of 14 independent FRAP experiments are shown. Bar, 1 μm. mGFP monomeric green fluorescent protein

a 5-kDa poly-L-lysine-coated slide glass to help recruit the complex to the two-dimensional (2D) surface. Under these conditions, the complex efficiently formed a readily distinguishable cylindrical self-assembly (Fig. 2c, Supplementary Fig. 2b, and Supplementary Movies 1 and 2). Quantifying signal intensities over time showed that most of these assemblies remained largely stable for up to 5 days, though a fraction (~10%) of them either showed a significant (~20%) reduction in signal intensities or appeared to collapse during the same period of time (Supplementary Fig. 2c). Measurement of the diameter of these assemblies ($n = 482$) revealed a negative exponential distribution pattern (Fig. 2d), suggesting that the self-assembly process occurs in a continuous-time stochastic fashion. The varying degrees of the cylindrical diameters are suggestive of the flexible character of the assembly. While the distribution pattern of assembly diameters remained similar under various concentrations (i.e., from 2 to 8 μM) of the Cep63 P1•Cep152 M4d complex (Supplementary Fig. 2d), the assemblies were rarely observed at 1.5 μM and were not detectable at 0.75 μM. As the self-assembly's diameter decreased, its height was reduced exponentially (Fig. 2e). It is possible that the self-assemblies with smaller diameters (i.e., smaller bases for cylindrical structures) are either physically unstable or structurally unfavored during the assembly process. Notably, the diameter of the mGFP fluorescence in the assembly was somewhat larger than that of the mCherry fluorescence (Fig. 2f, g), hinting that the self-assembly was generated in an mGFP-Cep152 M4d-outward fashion. Consistent with the requirement of the hydrophobic motifs in forming the fast-sedimenting Cep63 P1•Cep152 M4d species (Fig. 2a), the Cep63 P1(4A)•Cep152 M4d(5A) hydrophobic mutant failed to generate any detectable self-assemblies (Supplementary Fig. 2b).

To confirm the self-assembling activity of the Cep63 P1•Cep152 M4d complex described above, we examined whether mCherry-Cep63 FL and mGFP-Cep152 M6(924–1295) (the longest Cep152 form that we successfully purified) can form self-assemblies under the same conditions described in Fig. 2c. Albeit very infrequently, mCherry-Cep63 generated a cylindrical self-assembly by itself, whereas mGFP-Cep152 M6 alone did not (Supplementary Fig. 2e). In the presence of both proteins, however, the number of discernable self-assemblies was greatly increased. In addition, these assemblies exhibited a directional organization—a Cep63-in and Cep152-out morphology—with an mCherry-mGFP interdistance of 165–275 nm (Fig. 2g and Supplementary Fig. 2e–h). Remarkably, self-assemblies generated with Cep63-mCherry (i.e., mCherry is fused at the C terminus of Cep63) and mGFP-Cep152 M6 exhibited a significantly diminished mCherry-mGFP interdistance (Fig. 2g and Supplementary

Fig. 2i), suggesting that the N terminus of Cep63 is placed away from the N terminus of Cep152. Collectively, these observations are consistent with the view that the Cep63•Cep152 complex is radially arranged from the axis of the cylindrical assembly in a Cep63-in and Cep152-out fashion, with the Cep63 N terminus pointing inward and Cep152 N terminus pointing outward (see below). In line with this argument, analysis of the localization patterns of endogenous proteins revealed that Cep63 localizes to the inner face of Cep152 (Supplementary Fig. 2j). To further characterize the nature of this assembly below, we used the mCherry-Cep63 P1•mGFP-Cep152 M4d assemblies because of the difficulty of purifying Cep63 FL and Cep152 M6 proteins and an increased vulnerability of longer coiled-coil proteins to nonspecific local interactions due to their characteristic flat free energy landscape[21] (note that the cylinders made of longer forms tend to collapse towards the edge of the assembly in Supplementary Fig. 2f and Supplementary Movie 6).

**Dynamic nature of the Cep63–Cep152 self-assembly**. To observe the mechanochemical process of forming the Cep63 P1•Cep152 M4d self-assembly in real time, we carried out SIM-total internal reflection fluorescence (TIRF) time lapse immediately after placing the mCherry-Cep63 P1•mGFP-Cep152 M4d complex on a poly-L-lysine-coated glass bottom dish at RT ($t = 0$). Although the self-assemblies appeared at various times, ring-like assemblies with both mCherry and mGFP fluorescence were detectable as early as $t = $ ~50 min, and their signals steadily increased until $t = $ ~100 min after incubation (Fig. 2h and Supplementary Movie 3). Interestingly, the diameter of the Cep63 P1•Cep152 M4d self-assembly did not appear to change during the entire course of forming the architecture (Fig. 2i and Supplementary Fig. 2k), suggesting that it is determined early in the assembly process. Fluorescent recovery after photobleaching (FRAP) revealed that the fluorescence of the mCherry-Cep63 P1•mGFP-Cep152 M4d self-assembly was fully recovered in ~10 min after the bleach (Fig. 2j), suggesting that the self-assembly undergoes a dynamic exchange of its components with those in the surroundings. This observation was to some extent unexpected, considering that the self-assemblies appeared to be largely stable when placed under the assembly buffer conditions (Supplementary Fig. 2c). Given that self-assembly is a spontaneous free energy-minimization process ($\Delta G < 0$), reversing this process by overcoming the energy barrier for disintegrating a macroscale cylindrical architecture could be thermodynamically unfavorable and kinetically slow ($k_{off} \ll k_{on}$). To examine whether the dynamic exchange of the components occurs preferentially at the

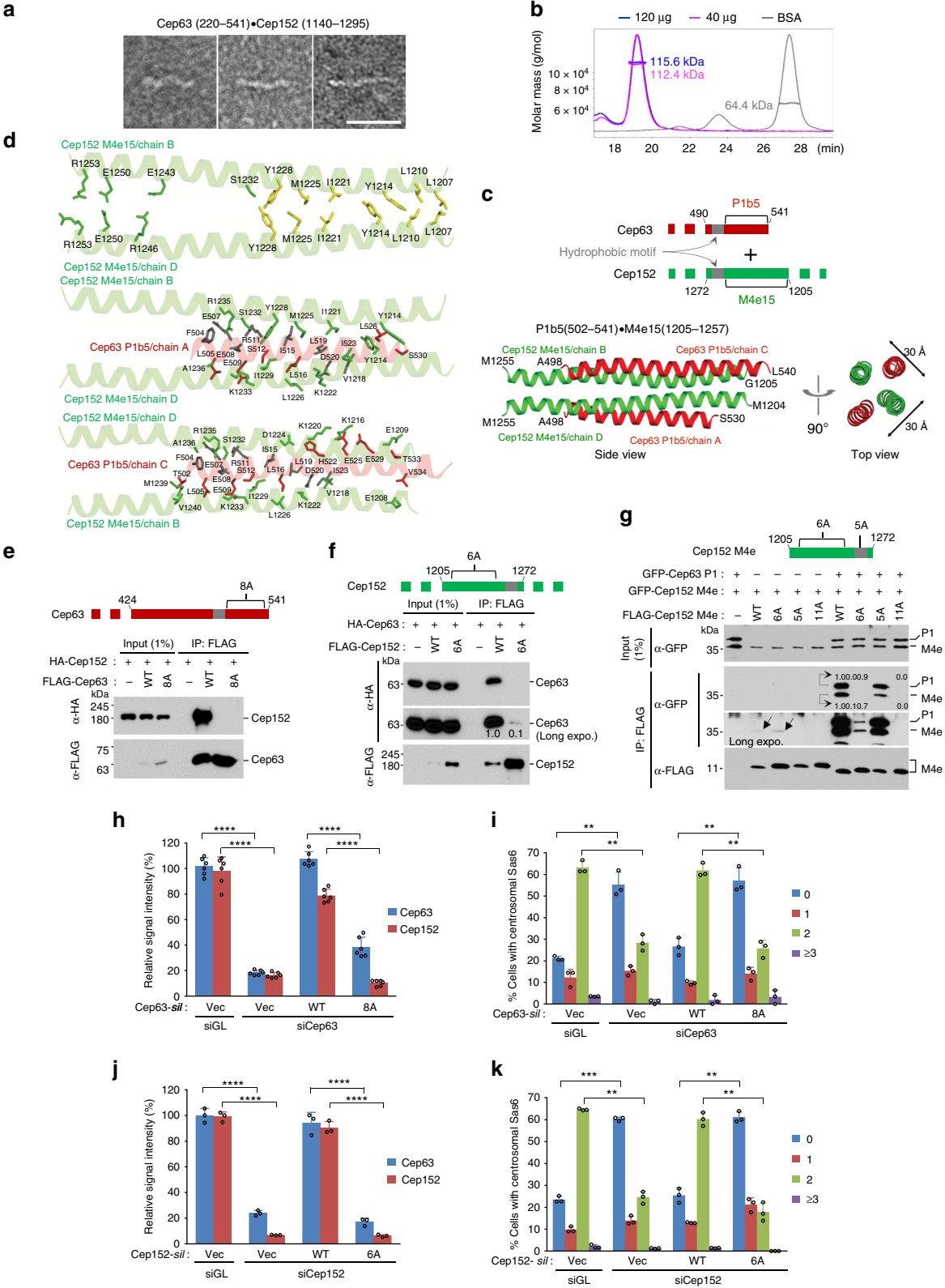

longitudinal ends of the cylindrical self-assembly, the Cep63 P1•Cep152 M4d self-assembly decorated with fluorescein iso-thiocyanate (FITC) was incubated with the mCherry-Cep63 P1•mGFP-Cep152 M4d complex for a short period of time and subjected to 3D-SIM analysis (Supplementary Fig. 2l). Notably, while the mCherry fluorescence was somewhat preferentially detected toward the ends of the cylinders, it was not strictly

confined to these regions (Supplementary Fig. 2m). In a related experiment, we observed that the fluorescence intensities of mCherry-Cep63 P1•mGFP-Cep152 M4d assemblies remain lar-gely unchanged under the conditions treated with buffer alone (as shown in Supplementary Fig. 2c), whereas they were nearly undetectable in the presence of the non-fluorescent Cep63 P1•Cep152 M4d complex (Supplementary Fig. 2m). This finding

**Fig. 3** Structural basis for forming the Cep63•Cep152 complex and its biological function. **a** Transmission electron micrograph showing the Cep63 (220–541)•Cep152(1140–1295) complex after negative staining. Bar, 30 nm. **b** Size-exclusion chromatography-multi-angle light scattering (SEC-MALS) analysis showing a stable heterotetrameric Cep63(220–541)•Cep152(1140–1295) complex. Bovine serum albumin (BSA) serves as a control. **c** Crystal structure of the Cep63 P1b5•Cep152 M4e15 complex forming an antiparallel four-helix bundle. The N and C termini of each protein are indicated. **d** Pairwise M4e15–M4e15 interactions (top) and P1b5 (A and C chains) interactions with M4e15 (B and D chains) (middle and bottom). The two P1b5 chains do not directly interact with each other. Side chains involved in the interaction are shown in the same color (i.e., green for M4e15 and red for P1b5) as the helices to which they belong. Side chains colored in yellow and gray indicate the residues mutated to 6A and 8A in Cep152 and Cep63, respectively. **e–g** Immunoprecipitation (IP) and immunoblotting analyses using transfected HEK293 cells. The schematic diagram illustrates the region where marked mutations are introduced. Gray boxes, hydrophobic motifs; numbers, relative signal intensities. Note that the 6A mutations greatly impair not only the central M4e–M4e interaction but also the P1–M4e interaction (**g**). **h–k** Analysis of U2OS cells stably expressing the indicated constructs after silencing for control luciferase (siGL), Cep63 (siCep63), or Cep152 (siCep152). The resulting cells were immunoblotted and immunostained (see Supplementary Fig. 3i–n). Relative fluorescence intensities of centrosome-associated Cep63 or Cep152 were quantified from three independent experiments (≥60 cells per experiment) (**h**, **j**). The number of centrosomal Sas6 dot signals per cell (classified as 0, 1, 2, and ≥ 3) was quantified from three independent experiments (≥200 cells per experiment) (**i**, **k**). Error bars, s.d.; $**P < 0.01$; $***P < 0.001$, $****P < 0.0001$ (unpaired two-tailed $t$ test)

suggests the exchange between the mCherry-Cep63 P1•mGFP-Cep152 M4d complex in the cylindrical self-assembly and the Cep63 P1•Cep152 M4d complex in solution can occur dynamically, likely because of a low energy barrier for this event, and further supports the FRAP results shown in Fig. 2j. The reversibility of the mCherry-Cep63 P1•mGFP-Cep152 M4d self-assembly is in sharp contrast to the network-like Cnn assembly that exhibited no turnover after photobleaching[11]. The dynamic nature of the self-assembly may be important to allow various centrosomal proteins to diffuse through the Cep63•Cep152 architecture.

**Molecular basis underlying the Cep63–Cep152 interaction**. To gain additional insights into the molecular nature of the Cep63•Cep152 self-assembly, we purified a longer Cep63 (220–541)•Cep152(1140–1295) (no mCherry or mGFP-tagged) complex and performed negative stain electron microscopy to visualize its overall morphology. Although varied somewhat in length, the complex exhibited an elongated morphology with a contour length of a mean of $31.04 \pm 4.11$ nm (s.d.) ($n = 30$) (Fig. 3a and Supplementary Fig. 3a). These findings hint that the Cep63•Cep152 self-assemblies observed in Fig. 2c and Supplementary Fig. 2e and 2f are made of a stretched rod-like structure. Similar to the Cep63 P1•Cep152 M4d complex (Fig. 2a), the longer complex also formed a 2:2 heterotetramer (expected MW 112.54 kDa) (Fig. 3b) capable of generating a faster-sedimenting species, matching to a predicted 4:4 heterooctameric complex (Supplementary Fig. 3b, arrow).

Next, we investigated how Cep63 and Cep152 generate a heterotetrameric complex that may serve as a building block for the assembly. Since Cep63 P1 and Cep152 M4d are sufficient to interact with each other and generate a self-assembly similar to that of the longer Cep63 FL•Cep152 M6 complex (Fig. 2c and Supplementary Fig. 2e, f), the activity that drives the Cep63•Cep152 self-assembly may originate entirely from the biochemical/biophysical property of the Cep63 P1•Cep152 M4d complex. However, our effort to determine the crystal structure of the P1•M4d complex was unsuccessful because of heavy precipitation caused by the hydrophobic motifs within P1 and M4d (Supplementary Fig. 1c, d). As expected, a smaller Cep63 P1b5 (502–541)•Cep152 M4e15(1205–1257) complex lacking the hydrophobic motifs maintained its 2:2 heterotetrameric state (expected MW 25.56 kDa) without forming any detectable high MW complex (Supplementary Fig. 3c). Size-exclusion chromatography-multi-angle light scattering (SEC-MALS) performed with decreasing concentrations of the complex revealed an apparent biphasic mode of MW changes, suggesting that the heterotetramer dissociates in two distinct steps with estimated $K_d$ values of ~117 μM and ~60 μM (Supplementary Fig. 3d).

## Table 1 Data collection and refinement statistics (molecular replacement)

| | Cep63 P1b5(502–541)•Cep152 M4e15 (1205–1257) |
|---|---|
| **Data collection**[a] | |
| Space group | $P2_12_12_1$ |
| Cell dimensions | |
| $a, b, c$ (Å) | 28.07, 41.52, 205.62 |
| $\alpha, \beta, \gamma$ (°) | 90.0, 90.0, 90.0 |
| Resolution (Å) | 19.87–2.50 (2.54–2.50)[b] |
| $R_{sym}$ or $R_{merge}$ | 0.083 (0.361) |
| $I/\sigma I$ | 16.68 (4.39) |
| Completeness (%) | 91.18 (73.92) |
| Redundancy | 6.9 (6.1) |
| **Refinement** | |
| Resolution (Å) | 19.87 - 2.50 |
| No. of reflections | 8163 |
| $R_{work}/R_{free}$ | 0.245/0.281 |
| No. of atoms | |
| Protein | 1489 |
| Ligand/ion | 0 |
| Water | 15 |
| $B$-factors (Å²) | 55.30 |
| Protein | 55.30 |
| Ligand/ion | 0 |
| Water | 54.00 |
| R.m.s. deviations | |
| Bond lengths (Å) | 0.004 |
| Bond angles (°) | 0.877 |

[a]The data set collected from single crystal
[b]Values within parentheses are for highest-resolution shell

Analyses of the crystal structures of the P1b5•M4e15 complex (PDB: 6CSU) (Table 1) and its related construct (PDB: 6CSV) (Supplementary Table 1) obtained at 2.5 Å resolution revealed that Cep63 and Cep152 form a four-α-helix bundle with tightly packed hydrophobic residues from each helix buried along the long axis of the heterotetrameric complex (Fig. 3c, d). In particular, identical hydrophobic residues in the parallelly aligned Cep152 helices appeared to constitute the backbone of the bundle by engaging in pairwise ladder-like interactions (Fig. 3d, top, and Supplementary Fig. 3e, f). Cep63 helices did not directly interact with each other, but rather interacted with both Cep152 helices in an antiparallel fashion (Fig. 3d, middle and bottom, and Supplementary Fig. 3e, f). These residues are highly conserved among vertebrates (Supplementary Fig. 1d). Unlike the recently determined *Drosophila* Cnn structure, whose helical bundles wound one another in a super-coil arrangement[11]

(Supplementary Fig. 3g), the four α-helices of the P1•M4d bundle show a virtually straight arrangement with periodicities of 3.70 residues per helical turn (Fig. 3c).

In support of the structural results in Fig. 3d, mutating eight conserved residues (F504A, E507A, E508A, R511A, I515A, L519A, D520A, I523A; gray residues in Fig. 3d, middle and bottom, and Supplementary Fig. 3e) found along the long interacting interface of Cep63 P1 abolished even the interactions between Cep63 FL and Cep152 FL (Fig. 3e). Strikingly, mutating the six residues (L1207A, L1210A, Y1214A, I1221A, M1225A, Y1228A; yellow residues in Fig. 3d, top, and Supplementary Fig. 3e) that gave rise to the ladder-like interactions between two parallelly arranged Cep152 M4e15 chains also greatly impaired the Cep63–Cep152 interaction (Fig. 3f). This is because the M4e15–M4e15 interaction serves as the platform for establishing the Cep63–Cep152 interaction (Fig. 3d, middle and bottom). In support of this finding and the cooperative interaction observed between P1 and M4d (Fig. 1c and Supplementary Fig. 1k), the 6A mutations disrupted not only the M4e–M4e interaction but also the M4e–P1 interaction under the conditions that allow the formation of a heteromeric P1•M4e complex (Fig. 3g). In addition, the 5A mutation notably diminished the level of the M4e–M4e interaction (Fig. 3g and Supplementary Fig. 3h), suggesting a role of the hydrophobic motif for the interaction (see below).

**The biological significance of the Cep63–Cep152 interaction.** To investigate the significance of the Cep63–Cep152 interaction for centriole duplication, we generated U2OS cells expressing the Cep63(8A) or Cep152(6A) mutants under the conditions where their respective endogenous proteins were silenced (Fig. 3h–k and Supplementary Fig. 3i–n). As expected, if Cep63 and Cep152 functioned as a complex, silencing either Cep63 or Cep152 was sufficient to diminish the level of both of these proteins recruited to centrosomes (Fig. 3h, j and Supplementary Fig. 3j, m), as reported previously[14,20,22]. However, another study suggests that Cep152 and Cep63 interact with distinct centriolar satellite proteins and that the recruitment of Cep63 to centrosomes depends on Cep152[29], prompting speculation that the establishment of the Cep63–Cep152 assembly is intricately regulated during cell-cycle progression. Interestingly, the expression of Cep63 or Cep152 completely restored centrosome-localized Cep63 or Cep152 fluorescence, whereas the Cep63(8A) or Cep152(6A) mutant defective in the cooperative Cep63–Cep152 interactions did not (Fig. 3h, j and Supplementary Fig. 3j, m). As a consequence of these defects, cells expressing Cep63(8A) or Cep152(6A) failed to recruit Sas6, a key element of the centriolar cartwheel structure[30], to centrosomes under the conditions where their respective wild-type (WT) can fully remedy the defect associated with Cep63 or Cep152 RNA interference (RNAi) (Fig. 3i, k and Supplementary Fig. 3k, n).

Next, since the hydrophobic motifs (gray box in Fig. 4a) adjacent to the tetrahelical Cep63 P1b5•Cep152 M4e15 bundle are critical for forming a higher-order self-assembly (Supplementary Fig. 2b, e) in vitro, we investigated whether they are required for proper function of Cep63 and Cep152 in vivo. Additional analysis showed that these motifs were present either adjoining or within the homomerizing region of P1(440–490) or M4e(1257–1272), respectively (Supplementary Fig. 4a–c). Thus, since the cooperative formation of the four-helix P1b5•M4e15 bundle is critical for the function of Cep63 and Cep152 (Fig. 3), these hydrophobic motifs may contribute to the Cep63–Cep152 interaction. Not surprisingly, the hydrophobic 4A and 5A mutations significantly impaired the Cep63–Cep152 interaction in an apparently additive fashion (Fig. 4b). Considering the

importance of the hydrophobic effect in clustering proteins[25,26], the hydrophobic motifs found in P1 and M4e (Supplementary Fig. 1c, d) may improve the proximity between Cep63 and Cep152 and thereby promote their interactions (Supplementary Fig. 4c). Consistent with the diminished level of the Cep63 (4A)–Cep152(5A) interaction (Fig. 4b), U2OS cells expressing either Cep63(4A) or Cep152(5A) under the conditions that deplete its endogenous protein exhibited a significantly (~40%) diminished level of centrosome-localized Cep63 and Cep152 fluorescence and failed to recruit Sas6 to centrosomes (Fig. 4c–f and Supplementary Fig. 4d–i). These observations hints that, in addition to mediating the Cep63–Cep152 interaction, the hydrophobic motifs may play an additional (e.g., architectural) role in the formation of a functional Cep63–Cep152 self-assembly. Because of the technical difficulty of generating U2OS cells expressing both Cep63(4A) and Cep152(5A) mutants under Cep63 and Cep152 double RNAi conditions, we could not assess the additive effect of Cep63(4A) and Cep152(5A) mutations.

**Functionality of the Cep63–Cep152 self-assembly.** Next, we investigated the ability of the Cep63•Cep152 self-assembly to recruit its downstream components, such as Plk4 and Sas6. Coimmunoprecipitation analyses showed that Cep63 associated with Plk4 via Cep152 (Supplementary Fig. 5a, b). Consistent with this finding, the N-terminal region of Cep152 (either the 1–217 (i.e., N217) fragment[31] or a minimal 1–70 (N70) fragment[32]) directly interacts with the cryptic polo-box of Plk4. Therefore, we purified mCherry-Cep63 P1 complexed with a chimeric mGFP-Cep152 N70-M4d(1205–1295) (we failed to purify a longer N217-M4d because it is highly insoluble), generated a self-assembly, and incubated it with purified Plk4 FL. Not surprisingly, the self-assemblies containing the mGFP-Cep152 N70-M4d, but not the parental mGFP-Cep152 M4d, effectively recruited Plk4 to the location where mGFP fluorescence tagged to the N terminus of Cep152 was detected (Fig. 5a, Supplementary Fig. 6, and Supplementary Movie 4).

In cultured U2OS cells, mCherry-Cep63 FL and mGFP-Cep152 FL efficiently localized to centrosomes (Supplementary Fig. 7a), suggesting that the tags fused to N termini did not alter the ability of these proteins to assemble at the endogenous centrosomal site. However, their overexpression led to the generation of macroscale spherical assemblies in cytosol with no distinguishable structural features (Supplementary Fig. 7b). Since the poly-L-lysine-coated coverslip promoted the formation of the Cep63 P1•Cep152 M4d self-assembly in vitro by attracting the complex to a 2D surface (Fig. 2c), we fused a 26-residue-long amphipathic stage V sporulation protein M (SpoVM) from *Bacillus subtilis*[33] to the N terminus of mCherry-Cep63 to target the protein onto the membrane surface (Supplementary Fig. 7c). Under these conditions, coexpressed SpoVM-mCherry-Cep63 P1 and mGFP-Cep152 M4d, but not the corresponding hydrophobic P1(4A) and M4d(5A) mutants, generated ectopic cylindrical assemblies (Supplementary Fig. 7d–g) with an overall morphology similar to that of the in vitro self-assemblies formed on the poly-L-lysine-coated surface (Fig. 2c). Like the in vitro self-assemblies, all of the in vivo assemblies exhibited an upright axial orientation with a Cep63-in and Cep152-out arrangement (Supplementary Fig. 7e–g and Supplementary Movie 7). Although the number of assemblies is low, SpoVM-mCherry-Cep63 FL and mGFP-Cep152 FL also generated ectopic cylindrical assemblies in a Cep152-outward fashion (Supplementary Fig. 7h).

To examine the ability of ectopically established Cep63•Cep152 self-assemblies to recruit downstream components, we coexpressed SpoVM-mCherry-Cep63 FL along with either Cep152 N217-fused M6(924–1295) or its parental N217-deficient form in

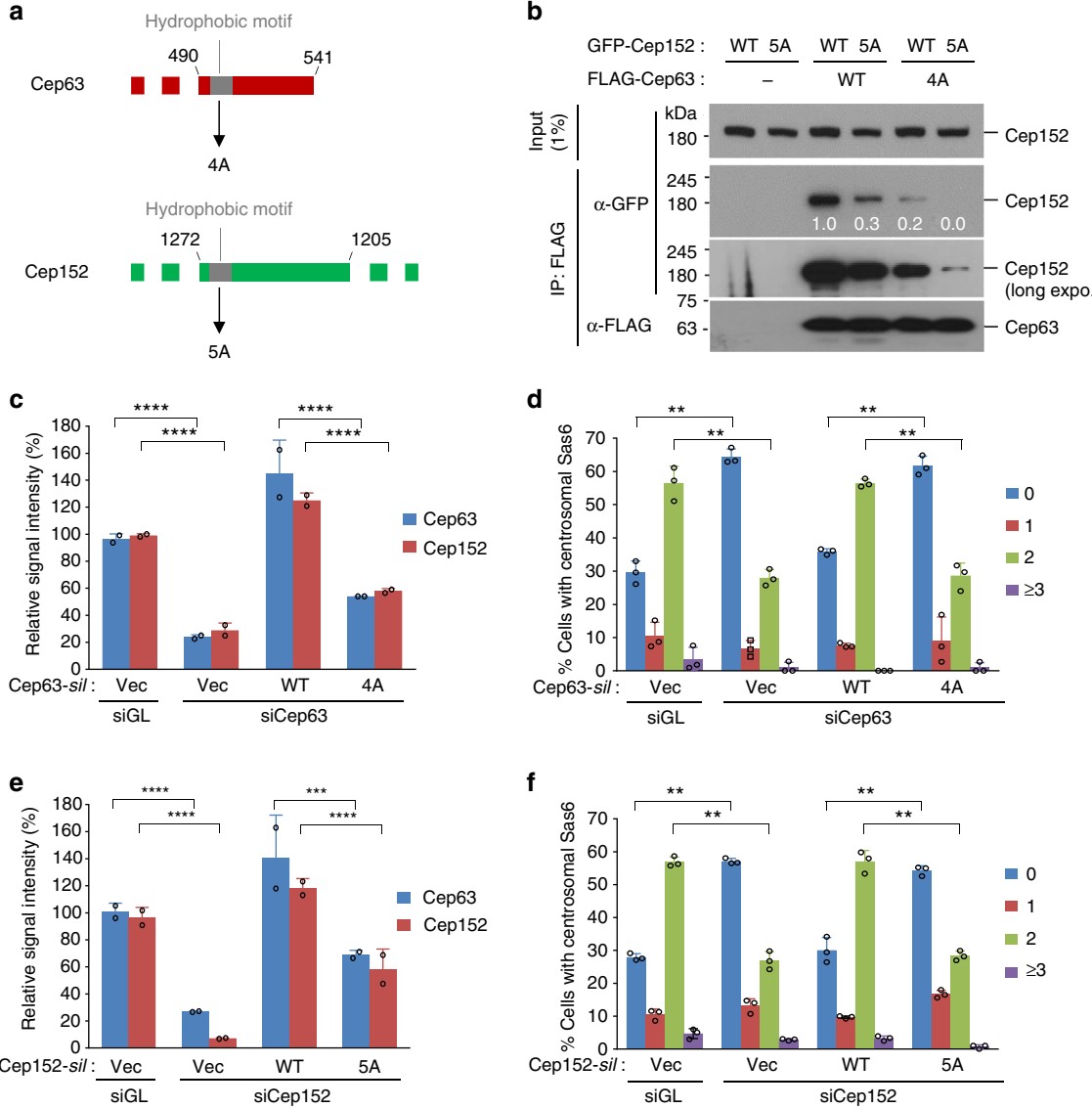

**Fig. 4** Hydrophobic motifs promote the localization and function of Cep63 and Cep152. **a** Schematic diagram illustrating the hydrophobic motifs (gray boxes) present in Cep63 P1b and Cep152 M4e. **b** Immunoprecipitation (IP) and immunoblotting analyses using transfected HEK293 cells. Numbers, relative signal intensities. **c**–**f** Analysis of U2OS cells stably expressing the indicated constructs after silencing for luciferase (siGL), Cep63 (siCep63), or Cep152 (siCep152). The resulting cells were immunoblotted and immunostained (see Supplementary Fig. 4d–i). Relative fluorescence intensities of centrosome-associated Cep63 or Cep152 were quantified from three independent experiments (≥70 cells per experiment) (**c**, **e**). The number of the centrosomal Sas6 per cell was quantified from three independent experiments (≥200 cells per experiment) (**d**, **f**). Error bars, s.d.; **P < 0.01; ***P < 0.001, ****P < 0.0001 (unpaired two-tailed $t$ test)

U2OS cells. As expected, the resulting self-assemblies containing the Plk4-binding N217 fragment[31], but not the N217-deficient control, effectively recruited coexpressed Plk4 to the outskirts of the Cep63•Cep152 N217-M6 assemblies (Supplementary Fig. 8a). Other studies have shown that Plk4-dependent phosphorylation of a centriole duplication factor, STIL[34–36], is required for the recruitment of Sas6[37–39]. Therefore, to further examine whether the SpoVM-mCherry-Cep63•Cep152 N217-M6 self-assemblies in Supplementary Fig. 8a can recruit Sas6, U2OS cells were additionally transfected with STIL and Sas6 then treated with okadaic acid, a protein phosphatase 2A inhibitor, to activate Plk4[40]. Not surprisingly, the Cep152 N217-M6-containing assemblies, but not the respective N217-deficient form, recruited both Plk4 and Sas6 on the surface of the Cep63•Cep152 self-assemblies (Fig. 5b, Supplementary Fig. 8b, and Supplementary Movie 5). Unlike Sas6 WT detected as a congregated dot- or

patch-like morphology, the oligomerization-defective Sas6 (F131E) mutant exhibited greatly weakened and unfocused signals (Supplementary Fig. 8c), as was previously observed[41]. Expression of either Cep63 P1 or Cep152 N70-M4d significantly diminished the fraction of cells with recruited Sas6 signals (Supplementary Fig. 8d), suggesting that, when expressed alone, the P1 or N70-M4d fragments can inhibit centriole duplication in a dominant-negative manner.

The in vivo data described above, together with the in vitro self-assembly in Fig. 2, suggest that the Cep63•Cep152 complex is radially arranged to form a cylindrical self-assembly in a Cep152-outward fashion (Fig. 5c). Since Cep63 P1 and Cep152 M4d form an elongated four-helix bundle (Fig. 3) that may serve as a building block for the self-assembly, the mode of interaction between the bundles may likely determine the directionality of the assembly that positions Cep152 at the outskirts of the Cep63

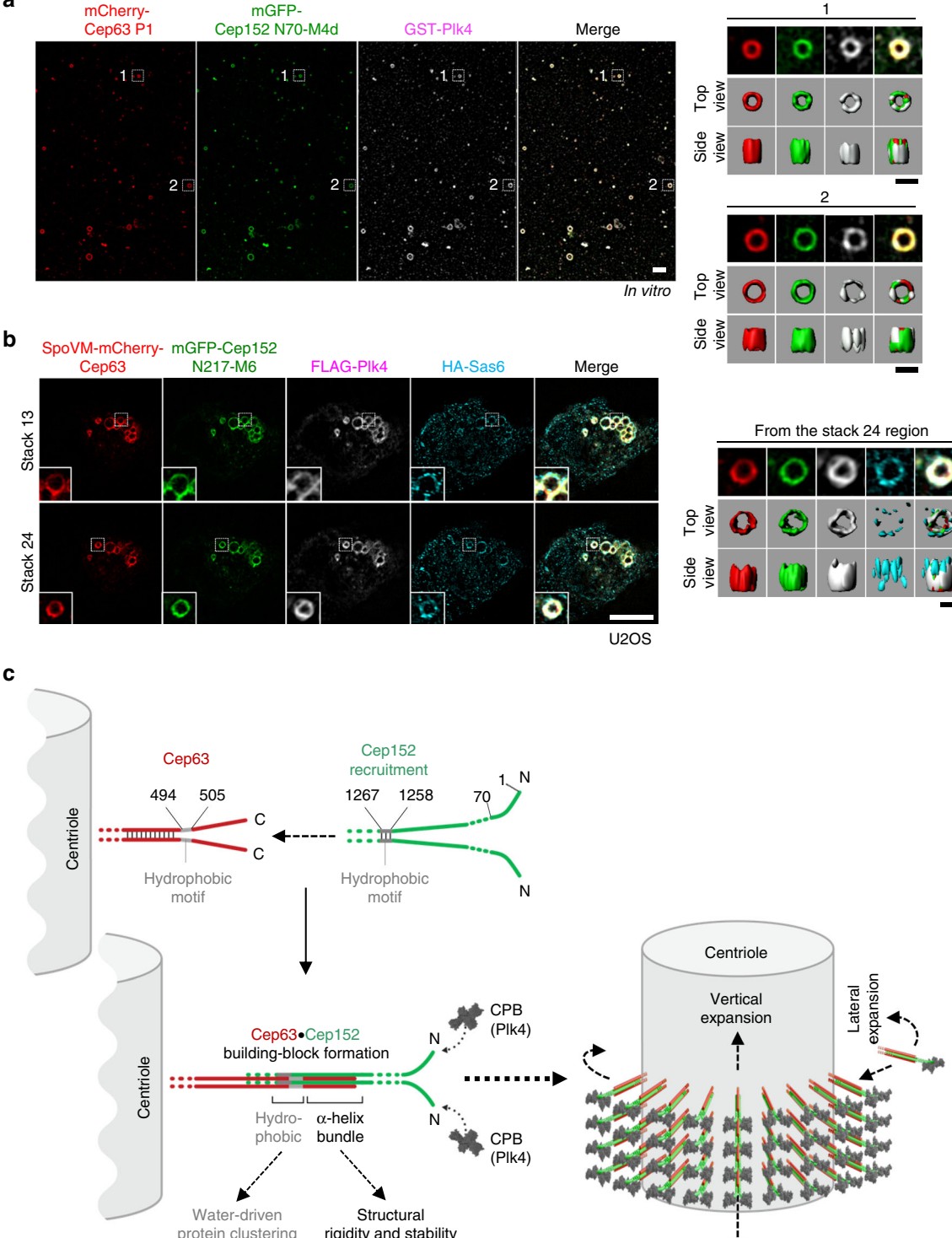

**Fig. 5** Recruitment of Plk4 and Sas6 by the Cep63-Cep152 self-assembly. **a** Three-dimensional structured-illumination microscopy (3D-SIM) analysis showing the ability of Cep63 P1•Cep152 N70-M4d self-assemblies to recruit Plk4 in an N70 (a minimal Plk4-binding motif)-dependent manner in vitro (see Methods for details). Surface rendering (right) is generated from the assemblies marked by dotted boxes. Bar, 1 μm; bar for rendered image, 0.5 μm. **b** 3D-SIM analysis of transfected U2OS cells showing the ability of Cep63•Cep152 N217-M6 self-assemblies to recruit Plk4 and Sas6 (see details in Methods). Surface rendering (right) from the assembly marked in Stack 24 (dotted box). Bar, 5 μm; bar for rendered image, 0.5 μm. **c** Model illustrating how Cep63 and Cep152 interact to form a heterotetrameric α-helical bundle and self-assemble into a higher-order cylindrical architecture at the proximal end of centrioles. The process of forming the four-helix bundle is explained in Supplementary Fig. 4c. We propose that the four-helix bundle functions in concert with the adjacent hydrophobic moiety to achieve the radial arrangement of the Cep63•Cep152 complex in the Cep152 N terminus-outward fashion (see Discussion for details). The resulting self-assembly appears to be a highly dynamic architecture, constantly exchanging its components with those in the surroundings

arrangement. Uninterrupted lateral and vertical interactions among the bundles may lead to the formation of a cylindrical higher-order assembly (see Discussions for details).

## Discussion

Macromolecular self-assembly is a fundamental intracellular process that underlies the construction of a higher-order structure in living organisms. However, except for some of the filamentous structures (e.g., actin and MTs) and mesoscale assemblies (e.g., clathrin-coated vesicles), little is known about how distinct protein subunits generate a building block complex that can assemble into a micron-scale three-dimensional architecture. Here we showed that two centrosomal scaffolds, Cep63 and Cep152, cooperatively interact with each other to generate a heterotetrameric α-helical bundle, which in turn self-assembles into a higher-order cylindrical architecture that resembles the localization morphology of endogenous Cep152 detected around a centriole[27]. Unlike actin or tubulins—which harness the energy from the hydrolysis of ATP or GTP, respectively, to generate filamentous polymers[42]—the entire assembly process occurs spontaneously without consuming any external energy.

Remarkably, short (~50 residues) coiled-coil heterotetramerization regions of Cep63 and Cep152 and their neighboring hydrophobic motifs (i.e., Cep63 P1 and Cep152 M4d) were necessary and sufficient to generate the cylindrical self-assembly capable of recruiting its guest molecules critical for centriole duplication in vivo. Since the hydrophobic effect can drive protein–protein interactions[25,26], we hypothesize that the hydrophobic motifs found in Cep63 P1 and Cep152 M4d play an important role not only in driving the Cep63–Cep152 interaction, as observed in Fig. 4b, but also in increasing the local concentration of the resulting heterotetrameric bundle to trigger self-assembly. In addition, the tightly structured helical bundle may provide structural rigidity and stability to the higher-order architecture generated by these two proteins (Fig. 5c).

The data provided here suggest that the heterotetrameric P1•M4d helical bundle serves as a building block that organizes into a higher-order architecture in such a way that its long axis is radially arranged around a centriole in a Cep152-outward fashion (Fig. 5c). The mechanism underlying the Cep63 P1–Cep152 M4d interaction may predetermine the orientation of the oppositely aligned P1 and M4d helices. The formation of a cylindrical higher-order assembly is likely the result of processive molecular recognitions that occur through the repetitive cycles of lateral and vertical interactions among building blocks. In this view, the intrinsic biochemical and biophysical properties of the P1•M4d complex may govern not only the spatial organization (i.e., unidirectional orientation and radial alignment) but also the geometrical arrangement (i.e., cylindrical shape of the assembly) of the Cep63•Cep152 scaffold around a centriole.

At present, the molecular basis of how Cep63 P1•Cep152 M4d tetrameric building blocks interact with one another to form a higher-order self-assembly is unknown. Further investigation on understanding the biochemical and biophysical nature of inter-building-block junctional interactions will be a key step in revealing the processes of generating the Cep63•Cep152 self-assembly. Given the diameters of Cep63•Cep152 assemblies are widely distributed, the inter-building-block interactions are likely flexible enough to generate varying degrees of curvatures for the assembly. The assembly's physical flexibility could be important in offering structural resilience against diverse mechanochemical processes constantly occurring at centrosomes.

Like Cep63 and Cep152, a large number of centrosomal scaffold proteins (in humans: Cep120, Cep57, Cep63, Cep152, CPAP, Cdk5Rap2, PCNT, etc.) contain multiple coiled-coil regions, and

the proteins are concentrically localized around a centriole in a highly organized fashion[3–5]. Whether these proteins possess the physicochemical property of forming a self-assembled structure is unknown. The molecular nature of the Cep63•Cep152 self-assembly described here may serve as a paradigm in investigating the structure and function of these proteins. Notably, various mutations in Cep63 and Cep152 are associated with the development of diverse human diseases, including cancer and microcephaly[8,43,44]. Therefore, further investigation into how these mutations alter the structure and function of the Cep63•Cep152 assembly may be important in decoding the rules of pericentriolar assemblies as well as gaining a holistic view of how a defect in centrosomal architecture contributes to the development of human disorders.

## Methods

**Plasmid construction**. To construct GFP-fused Cep63 FL (pKM4858), P1 (424–541) (pKM4861), P1a(424–492) (pKM4859), or P1b(490–541) (pKM4860), a *Pme*I–*Not*I fragment was inserted into the pCI-neo-GFP vector (pKM3828) digested by the corresponding enzymes. The deletion forms of pCI-neo-GFP-Cep63 P1b were generated similarly as above. The inserts are a *Pme*I–*Not*I fragment containing P1b1(490–517) (pKM5118), P1b2(502–529) (pKM5119), P1b3 (514–541) (pKM5120), P1b4(490–529) (pKM5186), or P1b5(502–541) (pKM5154).

For the construction of various GFP-fused Cep152 forms, an *Xho*I–*Sma*I fragment containing FL(1–1654) (pKM3841), M3(500–800) (pKM4518), M4 (790–1380) (pKM4519), M5(1370–1654) (pKM4520), M4a(790–1155) (pKM4844), M4b(1156–1380) (pKM4845), M4c(1156–1295) (pKM4880), M4d(1205–1295) (pKM4881), M4e(1205–1272) (pKM4962), M4f(1273–1295) (pKM4963), M4e6 (1205–1240) (pKM5112), M4e10(1205–1250) (pKM5135), M4e12(1225–1272) (pKM5137), M4e13(1205–1260) (pKM5138), or M4e14(1215–1272) (pKM5139) was subcloned into the pEGFP-C1 vector (Clontech Laboratories, Inc.) digested by the corresponding enzymes. For the generation of Cep152 translational variant 1 (tv1) (NCBI Accession: NP_001181927.1) constructs, an *Xho*I–*Sma*I fragment containing the cDNA of M4L(790–1436) (pKM4837), M4La(790–1199) (pKM4882), or M4Lb(1156–1436) (pKM4883) was generated by reverse transcription-polymerase chain reaction using SuperScript III RT (Invitrogen), and then cloned into the pEGFP-C1 vector similarly as above. The pEGFP-C1-Cep152 M1(1–217) (pKM3561) and M2(1–512) (pKM3562) were created by inserting a corresponding *Sal*I–*Sma*I fragment into the pEGFP-C1 vector digested by the same enzymes. Internal deletions, ΔM4e(Δ1205–1272) (pKM5012) and ΔM4d (Δ1205–1295) (pKM4952), were subcloned by inserting an *Eco*RI–*Sal*I fragment into the pEGFP-C1 vector digested by the corresponding enzymes. To generate pEGFP-C1-Cep152(5A) (L1260A, I1261A, L1263A, I1266A, L1267A) (pKM5471), the C terminus of the pEGFP-C1-Cep152 (pKM3841) construct was replaced by a fragment of Cep152(5A) PCR product using *Sal*I–*Eco*RI digestion.

To generate FLAG₃-tagged Cep63 FL (pKM2798), P1(424–541) (pKM5111), P1b(490–541) (pKM5010), N(1–220) (pKM5779), M(210–405) (pKM5780), C (400–541) (pKM5781), or ΔP1b(1–489) (pKM4988), a *Pme*I–*Not*I fragment of Cep63 PCR product was subcloned into the pCI-neo-FLAG₃ vector (pKM2795) digested by the corresponding enzymes. The pCI-neo-HA-Cep63 (pKM1235) construct was generated by inserting an *Eco*RV–*Not*I fragment of PCR product into the pCI-neo-HA vector (pKM1209) digested by *Sma*I and *Not*I. To generate pCI-neo-FLAG₃-Cep63(4A) (L497A, I500A, F504A, L505A) (pKM5650) and pCI-neo-FLAG₃-Cep63(8A) (F504A, E507A, E508A, R511A, I515A, L519A, D520A, I523A) (pKM5935), the C terminus of the pCI-neo-FLAG₃-Cep63 (pKM2798) construct was replaced by a fragment of Cep63(4A) or Cep63(8A) PCR product using *Eco*RV–*Not*I digestion, respectively.

To generate pCI-neo-FLAG₃-Cep152 (pKM2809), an *Xho*I–*Sma*I fragment of Cep152 PCR product was cloned into the pCI-neo-FLAG₃ vector digested by *Xho*I and *Sma*I. FLAG₃-Cep152 M4e(1205–1272) (pKM4987) and FLAG₃-Cep152ΔM4e (Δ1205–1272) (pKM5346) were generated by inserting a respective *Pme*I–*Not*I fragment into the pCI-neo-FLAG₃ vector digested by the corresponding enzymes. A FLAG-tagged Cep152 M4(790–1380) (pKM6080) or the respective 6A (L1207A, L1210A, Y1214A, I1221A, M1225A, Y1228A) mutant (pKM6081) was created by inserting a corresponding *Pme*I–*Xho*I fragment into the pCI-neo-FLAG₃ vector digested by the same enzymes. For the construction of HA-tagged Cep152 constructs, an *Xho*I–*Sma*I fragment containing Cep152 FL (pKM4235) from pCI-FLAG₃-Cep152 or a *Pme*I–*Not*I fragment containing M4e(1205–1272) (pKM5966) from pCI-FLAG₃-Cep152 M4e was cloned into the pCI-neo-HA vector digested by the corresponding enzymes. An HA-tagged Cep152 M4(790–1380) (pKM6078) and the respective 6A mutant (pKM6079) were generated by inserting a *Pme*I–*Xho*I fragment into the pCI-neo-HA vector. An HA-tagged Plk4 (pKM3855) or STIL (pKM4483) construct was generated by inserting a *Pme*I–*Not*I fragment containing the respective FL gene into the pCI-neo-HA vector. An HA-tagged SAS6 (pKM4381) construct was generated by inserting a *Cla*I–*Xba*I fragment

containing the FL gene into the pcDNA3.1-HA vector digested with corresponding enzymes.

To generate pCI-neo-FLAG$_3$-Cep152(6A) (L1207A, L1210A, Y1214A, I1221A, M1225A, Y1228A) (pKM5930) or pCI-neo-HA-Cep152(6A) (pKM6076), the C terminus of either the pCI-neo-FLAG$_3$-Cep152 (pKM2809) or the pCI-neo-HA-Cep152 (pKM4235) construct was replaced by a fragment of Cep152(6A) PCR product using SalI–NheI digestion. The Cep152 M4e constructs, containing the 6A (pKM6364), 5A (pKM6365), or the combined 11A (i.e., 6A + 5A) (pKM6363) mutations, were generated by cloning a PmeI–NotI fragment containing the respective mutations into the pCI-neo-FLAG$_3$ vector digested with corresponding enzymes. A FLAG-tagged Plk4 construct (pKM3445) was generated similarly as above by inserting a PmeI–NotI fragment into the pCI-neo-FLAG$_3$ vector.

To generate mCherry-fused Cep63 expression constructs, a pCI-neo-mCherry vector was first created by inserting an AscI–PmeI fragment containing an mCherry open reading frame (ORF) into the pCI-neo vector (Addgene). To construct pCI-neo-mCherry-Cep63 FL (pLM4916) and pCI-neo-mCherry-Cep63 P1 (pKM4986), a PmeI–NotI fragment containing the respective gene was cloned into the pCI-neo-mCherry vector (pKM4961). The pCI-neo-SpoVM-mCherry vector (pKM6781) was constructed by inserting an AscI fragment containing the entire (26 residues) SpoVM ORF into the pCI-neo-mCherry vector. SpoVM-mCherry-fused Cep63 FL WT (pKM6720), Cep63 P1 WT (pKM6780), and the Cep63 P1(4A) mutant (pKM6779) were all generated by inserting a respective PmeI–NotI fragment into the pCI-neo-SpoVM-mCherry vector digested with the same enzymes.

mGFP-fused Cep152 FL WT (pKM6005) was generated by cloning an AgeI–SmaI fragment containing an EGFP (A206K) (a monomeric mutation[28])-fused Cep152 fragment into the pEGFP-C1 digested with the same enzymes. The construction of mGFP-Cep152 M4d (pKM6782), mGFP-Cep152 M4d(5A) (pKM6783), and mGFP-Cep152 M6 (pKM6723) was done by inserting a respective BamHI–NotI fragment into the pcDNA3 vector (Addgene). The mGFP-Cep152 N217 (N-terminal 1–217)-M6 (pKM6722) was similarly constructed by inserting a BamHI–NotI fragment into the pcDNA3 vector. Construction of FLAG-Sas6 (pKM7058) and its assembly-defective F131E mutant (pKM7059) was carried out by inserting a SalI–NotI fragment into the pCI-neo-FLAG$_1$ vector (pKM2794) digested with XhoI and NotI.

For the lentiviral constructs expressing small interfering RNA (siRNA)-insensitive Cep152-sil WT (pKM5767) and Cep152(5A)-sil (pKM5768), an AscI–XhoI fragment containing Cep152-sil WT or 5A mutations was inserted into a pHR′.J-CMV-SV-puro vector (pKM2994) digested by AscI and SalI. The pHR′.J-CMV-Cep152(6A)-sil (pKM6029) construct was cloned by inserting an AscI–PmeI fragment containing Cep152(6A) mutations into a pHR′.J-CMV-SV-puro vector digested by the same enzymes. For the construction of siRNA-insensitive lentiviral Cep63 constructs, pHR′.J-CMV-FLAG$_3$-Cep63-sil WT (pKM5659), 4A (pKM5660), and 8A (pKM6006) were cloned by inserting the respective AscI-SalI fragment into a pHR′.J-CMV-SV-puro vector digested by the corresponding enzymes.

For bacterial expression, pRSFDuet-1-His-mGFP-Cep152 M6 (pKM6368) and its respective 5A mutant (pKM6459) were constructed by inserting a corresponding XhoI-NotI fragment into a pRSFDuet-His-mGFP (A206K) vector digested with SalI and NotI. The pRSFDuet-1 vector was purchased from Addgene. pETDuet-1-His-mCherry-Cep63 (pKM6066) and its respective Cep63(4A) mutant (pKM6700) were generated by inserting a corresponding BamHI-NotI fragment into the pETDuet-1 vector (Addgene) digested with the same enzyme.

To construct pETDuet-1-His-mGFP-Cep152 M4d+mCherry-Cep63 P1 (pKM6714) and its corresponding Cep152 M4d(5A)+Cep63 P1(4A) mutant (pKM6715), a BamHI-NotI fragment containing mGFP-Cep152 M4d or its respective 5A mutant and a BglII-XhoI fragment containing mCherry-Cep63 P1 or its respective 4A mutant were individually cloned into the corresponding multiple cloning sites of the pETDuet-1 vector. pETDuet-1-His-mGFP-Cep152 N70-M4d+mCherry-Cep63 P1 (pKM6829) was created by inserting an XhoI-SalI fragment containing Plk4-binding Cep152 N70 into pKM6714 digested with SalI. pETDuet-1-His-Cep63 P1(424–541)+Cep152 M4d(1205–1295) (pKM5615) and its respective P1(4A) and M4d(5A)-containing mutant (pKM5616) were generated by inserting the BamHI-NotI fragment of P1 or P1(4A) and the NdeI-XhoI fragment of M4d or M4d(5A) into the corresponding sites in the pETDuet-1 vector. The construction of pETDuet-1-His-Cep63 P1b5(502–541)+Cep152 M4e15 (1205–1257) (pKM5500) and pETDuet-1-His-Cep63(220–541)+Cep152 (1140–1295) (pKM6018) was done by inserting a BamHI-NotI fragment of Cep63 P1b5(502–541) or Cep63(220–541) and an Nde-XhoI fragment of Cep152 M4e15 (1205–1257) or Cep152(1140–1295) into the pETDuet-1 vector digested with the corresponding enzymes.

For X-ray crystallization, pETDuet-1-His-MBP-tobacco etch virus (TEV) protease-Cep63 P1b5(502–541)+Cep152 M4e15(1205–1257) (pKM5359) was generated by cloning a PmeI-NotI fragment of P1b5 and an NdeI-XhoI fragment of M4e15 into the corresponding sites in the pETDuet-1 vector. To aid the structure determination of the Cep63 P1b5•Cep152 M4e15 complex described above, we also generated a fused construct (pKM5390) that links the P1b5(502–541) to M4e10 (1205–1250) through a five residue-long GGGSE linker. The construct was made by inserting an NdeI-XhoI PCR fragment into a pET28a-based His-MBP-TEV vector[45] published previously.

All the constructs and all the primers used in this study are listed in Supplementary Tables 2 and 3, respectively.

**Protein expression and purification**. All proteins for structural study were expressed in *E. coli* Rosetta strain (Novagen). To purify the proteins for X-ray crystallization, cells were cultured in Luria Broth medium at 37 °C until their optical density reaches to 0.8, and then proteins were expressed overnight at 18 °C with 0.5 mM isopropyl β-D-1-thiogalactopyranoside. To purify the Cep63 P1b5•Cep152 M4e15 complex (pKM5359), cells co-expressing His$_6$-MBP-fused P1b5 (502–541) and M4e15 (1205–1257) proteins were lysed in an ice-cold buffer [50 mM Tris-HCl (pH 7.5), 150 mM NaCl, and 5% (v/v) glycerol] by ultra-sonication and then centrifuged at $20,000 \times g$ for 40 min. The supernatant was applied to HisTrap HP column (GE Healthcare), and the bound protein was eluted with the lysis buffer with increasing concentrations of imidazole (20–500 mM). The protein was digested with a recombinant His$_6$-tagged TEV to cleave off the His$_6$-MBP tag from the target protein. The resulting protein was further purified by subjecting it to MBPTrap HP (GE Healthcare) and an additional HisTrap HP column to remove the His$_6$-MBP and His$_6$-TEV. The purified protein was subjected to HiLoad 16/60 Superdex 200 (GE Healthcare) SEC equilibrated with the final buffer [20 mM Tris-HCl (pH 7.5), 100 mM NaCl, and 0.5 mM TCEP]. Another recombinant protein, His$_6$-MBP-fused P1b5(502–541)-GGGSE-M4e10 (1205–1250) (pKM5390), was purified by the same procedure and its seleno-methionine substituted protein was used for phasing. All purified proteins showed more than 95% purity in sodium dodecyl-polyacrylamide gel electrophoresis analysis.

For the purification of Cep63 P1•Cep152 M4d (pKM5619), Cep63 P1(4A) •Cep152 M4d(5A) (pKM5620), Cep63 P1b5(502–541)•Cep152 M4e15 (1205–1257) (pKM5500), mCherry-Cep63 P1•mGFP-Cep152 M4d (pKM6714), mCherry-Cep63 P1•mGFP-Cep152 M4d (pKM6715), and Cep63(220–541) •Cep152(1140–1295) (pKM6018) complexes, cells expressing each of the target protein complexes were lysed in an ice-cold buffer [20 mM Tris-HCl (pH 7.5), 150 mM NaCl, and 5% (v/v) glycerol, 0.5 mM TCEP] by ultrasonication and then centrifuged at $40,000 \times g$ for 20 min. The supernatant was applied to HisTrap HP column (GE Healthcare), and the bound protein was eluted with the lysis buffer containing 500 mM imidazole. The eluted protein was subjected to HiLoad 16/60 Superdex 200 (GE Healthcare) SEC equilibrated with the final buffer [20 mM Tris-HCl (pH 8.0), 150 mM NaCl, and 0.5 mM TCEP].

For the purification of His-mCherry-Cep63 (pKM6066), His-mCherry-Cep63 (4A) (pKM6700), mGFP-M6 (pKM6368), or mGFP-M6(5A) (pKM6459), cells expressing the protein of interest were lysed in an ice-cold buffer [20 mM Tris-HCl (pH 7.5), 150 mM NaCl, 10% (v/v) glycerol, 0.5 mM TCEP] by ultrasonication and then centrifuged at $40,000 \times g$ for 20 min. The supernatant was applied to HisTrap HP column (GE Healthcare), and the bound protein was eluted with the lysis buffer with increasing concentrations of imidazole (20–300 mM). The eluted protein was dialyzed with a low-salt binding buffer [20 mM Tris-HCl (pH 7.5), 20 mM NaCl, 10% (v/v) glycerol, 0.5 mM TCEP] for more than 4 h and subjected to HiTrap Q HP (GE Healthcare) anion exchange chromatography. The bound protein was eluted with increasing concentrations of NaCl (20–500 mM). The protein eluted from ion exchange chromatography was subjected to HiLoad 16/60 Superdex 200 (GE Healthcare) SEC equilibrated with the final buffer [20 mM Tris-HCl (pH 8.0), 150 mM NaCl, and 0.5 mM TCEP].

**Sedimentation velocity analytical ultracentrifugation**. Bacterially expressed proteins (pKM5619, pKM5620, and pKM5500) were purified in the buffer indicated and diluted to various concentrations. Sedimentation velocity experiments were conducted at 50,000 rpm using an An-50 Ti Rotor ($200,000 \times g$ at 7.156 cm) on a Beckman Coulter ProteomeLab XL-I analytical ultracentrifuge following standard protocols[46]. Experiments on the Cep63 P1•Cep152 M4d complex and corresponding P1(4A)•M4d(5A) mutant were carried out at 4 °C to avoid non-specific aggregation. Stock solutions of the complexes in 20 mM Tris (pH 7.5), 500 mM NaCl, 10 mM 2-mercaptoethanol, 0.5 mM TCEP, and 5% (v/v) glycerol along with a matching buffer were used to prepare a series of sample solutions at concentrations ranging from 10 to 500 μM. Samples, kept continuously on ice, were loaded into pre-chilled, two-channel, sector-shaped cells in a cold room at 4 °C, and thermally equilibrated at zero speed in a pre-chilled analytical ultracentrifuge and rotor. Absorbance (280 nm) and interference velocity scans were subsequently acquired at ~5-min intervals. Time corrected data were analyzed in SEDFIT 15.01c[47] in terms of a continuous $c(s)$ distribution of sedimenting species using a maximum entropy regularization confidence interval of 0.68. Due to the presence of glycerol, interference data were only analyzed for the most dilute solution. The partial specific volumes of the Cep63•Cep152 complexes were determined based on their amino acid composition using SEDNTERP[48]. The solvent density and viscosity were determined experimentally at 20 °C on an Anton Paar DMA 5000 density meter and Anton Paar AMVn rolling ball viscometer, respectively, and corrected to values at 4 °C. Sedimentation coefficients were corrected to standard conditions in water at 20 °C, $s_{20,w}$. Experiments on the Cep63 P1•Cep152 M4e15 complex were carried out at 20 °C in a similar manner. Stock solutions of the complex in 20 mM Tris (pH 7.5), 100 mM NaCl, and 0.5 mM TCEP along with a matching buffer were used to prepare a series of sample solutions at concentrations ranging from 95 to 530 μM. The $c(s)$ analyses in SEDFIT returned distributions with excellent data fits. In this case, the solvent density and viscosity were calculated in SEDNTERP.

**SEC and MALS**. The Cep63 P1•Cep152 M4d complex was analyzed by SEC-MALS at 4 °C using an in-line high-performance liquid chromatography (Agilent Technologies 1260 Infinity), Superdex 200 Increase 10/300 GL (GE Healthcare), and MALS system (Wyatt DAWN HELEOS II and OPTILAB T-rEX). Two different amounts (40 and 120 µg) were loaded onto the SEC column in the final buffer [20 mM Tris-HCl (pH 8.0), 150 mM NaCl, and 0.5 mM TCEP] at the flow rate of 0.5 ml/min for 60 min. Data collection and analyses were carried out using Astra chromatography software (Wyatt, version 7.1.1.3). As a reference, 70 µg of bovine serum albumin was analyzed under the same conditions.

**Circular dichroism**. Experiments were conducted on a Jasco J-715 spectro-polarimeter (Jasco International Co., Japan). Spectra were acquired at 4 °C from 195 to 250 nm with the average of three readings using a 0.1 cm pathlength cuvette at a spectral bandwidth of 1 nm, data pitch of 1 nm, and a response time of 1 s. Scans of phosphate-buffered saline (PBS) solution were subtracted from each protein data. Proteins were scanned at a concentration of 0.1 mg/ml in PBS. The spectra were converted to mean residue ellipticity.

**Crystallization and data collection**. For crystallizations, the P1b5•M4e15 complex (pKM5359) or the P1b5-GGGSE-M4e10-linked form (pKM5390) was concentrated to 10mg/ml in a buffer containing 20 mM Tris-HCl, pH 7.5, 100 mM NaCl, and 0.5 mM TCEP. Crystals of the P1b5•M4e15 complex were grown at 20 °C with a hanging-drop vapor-diffusion method by mixing 1.5 µl of protein at a ratio of 1:1 with reservoir solution containing 0.1 M MES, pH 5.4, 23% (w/v) PEG 3350, and 0.2 M ammonium sulfate. Suitable crystals of the P1b5-GGGSE-M4e10 were grown with a hanging-drop vapor-diffusion method at 20 °C using reservoir solution containing 0.1 M Bis-Tris, pH 5.5, and 10% (w/v) PEG 3350.

Crystals were flash-frozen in liquid nitrogen after soaking in a reservoir solution including additional 25% (v/v) glycerol as a cryo-protector. The diffraction data sets were collected at 100 K on beamlines 22-ID and 22-BM of the Advanced Photon Source, Argonne, Illinois, USA. Then, the data sets were indexed, integrated, and scaled with HKL-2000[49].

**Structure determination**. The crystal structure of the P1b5-GGGSE-M4e10 linker form (6CSV; Supplementary Table 1) was solved by single-wavelength anomalous diffraction at an Se peak wavelength of 0.9793Å. The AutoSol and Hybrid Sub-structure Search of PHENIX software[50] were used for phasing and density modification. Missing residues were manually built based on an electron-density map using WinCoot[51]. Further refinement was carried out with refmac5 from CCP4 suite[52,53] and phenix.refine from PHENIX software[54].

To determine the crystal structure of the P1b5•M4e15 complex (6CSU; Table 1), molecular replacement was carried out with Phaser-MR in PHENIX, using the refined model of the P1b5-GGGSE-M4e10 linker form obtained above as an initial search model. Resulting model was improved manually using WinCoot and refined with refmac5 and phenix.refine[54]. During refinement, 5% randomly selected reflections excluded from refinement were used for free R-factor calculations to validate the model after refinement. Water molecules were added either automatically or manually using WinCoot. Data collection and refinement statistics are provided in Table 1 and Supplementary Table 1. The Ramachandran plots have 98.85, 0.86, and 0.29% of the residues in the favored, allowed, and outlier regions, respectively, for P1b5-GGGSE-M4e10 and 99.42, 0.58, and 0% of the residues in the favored, allowed, and outlier regions, respectively, for the Cep63 P1b5•Cep152 M4e15 complex.

**In vitro self-assemblies**. For the in vitro self-assemblies on the surface of slide glass (Thermo Fisher Scientific) and coverslip dish (No. 1.5, 35 mM dish; MatTek Corporation), the surface was treated with 100% methanol for 2 min, washed with water five times, and then additionally treated with 1 mg/ml poly-L-lysine (1–5kDa poly-L-lysine, Sigma-Aldrich) for 10 min. After washing the surface with an assembly buffer [20 mM Tris-HCl (pH 8.0), 150 mM NaCl, 0.5 mM TCEP] five times, 10 µl of protein samples (6 µM) was placed on the treated surface and incubated overnight in a humidified dark chamber at 4 °C. After washing the resulting sample twice with the assembly buffer, the assemblies on the slide glass were embedded in SlowFade (Thermo Fisher Scientific) for SIM. The assemblies on the coverslip dish were placed under the assembly buffer and then subjected to SIM-TIRF time-lapse and FRAP.

**Labeling in vitro self-assemblies with FITC isomer I**. To decorate Cep63 P1•Cep152 M4d self-assemblies with FITC in Supplementary Fig. 2l, 10 µl of self-assemblies generated on a coverslip were incubated with 240 µM FITC in the assembly buffer for 1 h at RT. The resulting sample was used for further experiment after washing out unreacted FITC in solution.

**Plk4 recruitment assay**. To examine the ability of in vitro-generated Cep63•Cep152 self-assemblies to recruit Plk4, the assemblies generated with the mCherry-Cep63 P1•mGFP-Cep152 N70-M4d complex (N70 of Cep152 contains a minimal Plk4-binding motif[31]) were treated with GST-fused Plk4 FL (10 ng/µl) purified from Sf9 cells (Sigma-Aldrich) for 2 h at 4 °C. Afterwards, the sample was stained

with anti-Plk4 antibody for 2 h at RT, which was subsequently decorated with an Alexa Fluor 647 anti-rabbit antibody for 1 h. Samples were mounted with SlowFade Antifade (Thermo Fisher Scientific) and then observed under 3D-SIM.

To examine the ability of Cep63•Cep152 self-assemblies to recruit Plk4 in vivo, U2OS cells were cotransfected with mCherry-Cep63 (pKM6720), mGFP-Cep152 N217-M6 (pKM6722; N217 contains a longer Plk4-binding motif), FLAG-Plk4 (pKM3445), HA-STIL (pKM4483), and HA-Sas6 (pKM4381). Ten hours post-transfection, cells were treated with 200 nM of okadaic acid for 2h to activate Plk4 (this step is important to induce Plk4-dependent STIL phosphorylation and subsequent Sas6 recruitment[37]), and then fixed with 4% paraformaldehyde. The resulting cells were then treated with 0.1% Triton X-100 for 5 min at RT and subjected to immunostaining analysis. Immunostained samples were mounted with ProLong Gold Antifade (Thermo Fisher Scientific) and then observed under SIM.

**3D SIM and surface rendering**. Images for in vitro self-assemblies were acquired using a Zeiss ELYRA S1 super-resolution microscope (Zeiss) equipped with an Alpha Plan-Apo ×63/1.46 oil objective, 405 nm/488 nm/561 nm/640 nm laser illumination, and standard excitation and emission filter sets. Acquired images were processed using the ZEN black software (Zeiss). To quantify time-dependent 3D-SIM signal intensities in Supplementary Fig. 2c, images were acquired under the same laser power and exposure time, and processed using raw scale. The diameter of the self-assembled cylinders was determined by measuring the longest peak-to-peak distance for a given toroid. To measure the diameter of the assemblies in Fig. 2d, we used SIM-converted images that allowed us to achieve the inter-signal distance of up to 33 nm/pixel. To obtain the quantified data provided in Supplementary Fig. 2g, h, images were acquired by a home-built instant linear SIM (iSIM) with the inter-signal distance of 55 nm/pixel (Hari Shroff, National Institute of Biomedical Imaging and Bioengineering, NIH) and analyzed by the ImageJ software. The height of the cylindrical assemblies in Fig. 2e were determined by counting the number of the z-stacks (110 nm/stack) covering the entire height of the signals emanating from the assembly. Due to the intrinsic mechanical limit of the ELYRA SIM microscope, all the height estimates have an error range of ±110 nm.

Where appropriate, SIM images were subjected to 3D surface rendering by using the Imaris software version 8.4.1 (Bitplane).

**SIM-TIRF time-lapse microscopy**. To achieve a high-resolution time-lapse imaging analysis by eliminating unassembled mCherry-Cep63 P1 and mGFP-Cep152 M4d signals in solution, a SIM-TIRF method that confines the structured illumination pattern to the TIRF plane was used. After placing 6 µM of the mCherry-Cep63 P1•mGFP-Cep152 M4d complex onto a poly-L-lysine-coated coverslip dish ($t = 0$), images were captured with a GE DeltaVision OMX SR microscope (SIM-TIRF mode) using Olympus PlanApo N ×60/1.42 oil PSF with Laser liquid™ ($n = 1.5200$). Images were processed by using the SoftWoRx 6.5.2 software (GE Healthcare). To determine the changes in fluorescent signal intensities over time, the intensities of fluorescent signals at each time point were determined after subtracting background signal intensities using the ImageJ version 1.51r software.

**Fluorescent recovery after photobleaching**. The fluorescent signals of self-assemblies generated overnight were bleached within a small region of interest using a laser at the 405nm wavelength (GE DeltaVision OMX SR microscope; conventional mode). Images were then captured with the SIM-TIRF mode at 5-s intervals, then processed using the same method used for the SIM-TIRF described above.

**Negative staining transmission electron microscopy**. To visualize the Cep63 (220–541)•Cep152(1140–1295) complex by negative stain electron microscopy, 10 ng/µl of the complex in a physiological buffer [20 mM Tris-HCl (pH 7.5), 150 mM NaCl, 0.5 mM TCEP] was loaded onto a negatively charged carbon grid for 10 s, washed with 1% uranyl formate (UF) twice, stained with fresh UF solution for 30 s, and then dried at RT. Images were acquired using the Hitachi H-7650 and the JEOL 1400 transmission electron microscope.

**Cell culture and transfection**. U2OS and HEK293 cells were cultured as recommended by the American Type Culture Collection (ATCC). Transfection was carried out using either Lipofectamine 2000 (Invitrogen) or polyethylenimine for protein overexpression, or Lipofectamine RNAiMAX (Invitrogen) for siRNA transfection. To effectively knockdown endogenous Cep63 or Cep152 proteins, U2OS cells were silenced twice using respective siRNAs (Supplementary Table 4) against control Luciferase (siGL), Cep63 (siCep63), or Cep152 (siCep152) for 2–3 days.

**Lentivirus production and generation of stable cell lines**. For generating lentiviruses expressing the gene of interest, HEK293 cells were cotransfected with pHR′-CMVΔR8.2Δvpr, pHR′-CMV-VSV-G (protein G of vesicular stomatitis virus), and pHR′.J-CMV-SV-puro-based constructs containing a respective gene, using the calcium phosphate coprecipitation of DNA with calcium chloride in a buffered saline/phosphate solution[55]. U2OS cells were infected and selected with 2

µg/ml of puromycin (Sigma-Aldrich). The resulting U2OS stable cells were transfected with appropriate siRNAs to knockdown the respective endogenous proteins (Supplementary Table 4).

**Immunoprecipitation and immunoblotting analyses**. Immunoprecipitation was carried out after lysing transfected cells in TBSN buffer [20 mM Tris-HCl (pH 8.0), 150 mM NaCl, 0.5% Nonidet P-40, 5 mM EGTA, 1.5 mM EDTA][56] in the presence of protease inhibitor cocktail (Roche). Supernatants obtained from centrifugation at $15,000 \times g$ for 15 min were incubated with anti-FLAG or anti-GFP antibody for 1 h and then further incubated with protein A/G-agarose (Santa Cruz Biotechnology) for 1 h before precipitation. Immunoblotting analysis was performed according to standard procedures using enhanced chemiluminescence substrate (Pierce). Signal intensities were quantified using the ImageJ software. All the antibodies used in this study are listed in Supplementary Table 5.

**Immunostaining and confocal microscopy**. Cells grown on coverslips were fixed with either 4% paraformaldehyde or 100% methanol, stained with the indicated primary antibodies (Supplementary Table 5), and appropriate Alexa fluorophore-conjugated secondary antibodies (Invitrogen). The resulting coverslips were mounted with ProLong Gold Antifade (Thermo Fisher Scientific) before analysis. Confocal images were acquired using the Zeiss LSM 780 microscope under the same laser intensity, and the maximum intensity projection of z-stack images was used to quantify fluorescence signal intensities using the Zeiss ZEN software version 2.1 (Carl Zeiss).

**TopFluor PC staining**. Asynchronously growing U2OS cells transfected with either mCherry control (pKM4961) or SpoVM-mCherry (pKM6781) were subjected to TopFluor phosphatidylcholine (PC) staining[57]. Briefly, 14 h after transfection, cells were fixed with 4% paraformaldehyde for 30 min at 4 °C. The resulting cells were treated with 10 µM of TopFluor PC for 10 min at RT, washed three times with PBS, and then observed after mounting the sample with ProLong Gold Antifade (Thermo Fisher Scientific).

**Statistical analysis**. All the experiments were performed at least three times independently. All values are given as mean of $n \pm$ s.d. $P$ values were calculated by unpaired two-tailed $t$ test from the mean data of each group.

## Data availability

Data supporting the findings of this manuscript are available from the corresponding author upon reasonable request. A reporting summary for this Article is available as a Supplementary Information file. The coordinates and structure factors of the Cep63 P1b5•Cep152 M4e15 complex (6CSU) and the Cep63 P1b5-GGGSE-Cep152 M4e10 complex (6CSV) have been deposited in the Protein Data Bank.

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

## Acknowledgements
We are grateful to Louis (Chip) Dye at the Eunice Kennedy Shriver National Institute of Child Health and Human Development, as well as Ulrich Baxa, Kunio Nagashima, Natalia de Val, and Kedar Narayan at the National Cancer Institute, for their assistance with electron microscopy; Hari Shroff at the National Institute of Biomedical Imaging and Bioengineering for iSIM imaging; Xufeng Wu and Grzegorz Piszczek at the National Heart, Lung, and Blood Institute for SIM-TIRF imaging and circular dichroism analyses, respectively; Yanling Liu at Frederick National Laboratory for Cancer Research for 3D surface-rendered movies; Byung Ha Oh at KAIST for structural insights; and Alex Mogilner at New York University and José Faraldo-Gómez at the National Heart, Lung, and Blood Institute for offering their biophysical insights into the cylindrical self-assembly. We also thank Yaozong Chen and Raymond L. Erikson for a critical reading of the manuscript. This research was supported by the Intramural Research Program of the National Institutes of Health (National Cancer Institute (K.S.L.) and the National Institute of Diabetes and Digestive and Kidney Diseases (R.G.)); the R&D Convergence Program (CAP-16-03-KRIBB) of National Research Council of Science & Technology (NST) of South Korea and a KRIBB Research Initiative Program (B.Y.K); a NST grant CAP-16-03-KRIBB, (J.K.B.); and a Korean Biomedical Scientist Fellowship from the KRIBB Research Initiative Program (E.L).

## Author contributions
T.-S.K., L.Z., J.I.A., L.M., J.M.L., J.-E.P., J.K.B., B.Y.K. and K.S.L. designed and performed Cep63–Cep152 interaction and self-assembly experiments; Y.C., E.L., L.F. and Y.-X.W. carried out structural analysis; R.G. performed sedimentation velocity analysis; L.Z. conducted SIM-TIRF; L.M. and J.-E.P performed negative-staining transmission electron microscopy; and K.S.L., T.-S.K., L.Z., J.I.A., Y.C., E.L. and R.G. wrote the manuscript.

## Additional information

**Competing interests:** The authors declare no competing interests.

