## [Peer Review File · Nature Communications]

Reviewers' Comments:

Reviewer #1:

Remarks to the Author:

Kim et al. describe that in vitro and in vivo a complex of CEP63 and CEP152 fragments is able to cooperatively form cylindrical assemblies that resemble the observed CEP63 and CEP152 localisation around centrioles in cells. The authors complement these findings by providing a high-resolution structural description of a critical protein-protein interaction interface involved in this process and show that this interface is essential for centriole duplication. Their structural work also suggests models for the exact organisation of CEP63 and CEP152 assemblies around centrioles that is overall in agreement with previous localisation studies with both proteins in cells.

This work is potentially an important contribution towards understanding how the assembly of such a large and complex protein assembly as the centriole can arise from the scaffold forming / self-assembly properties of a few of its core constituents. As such this work would be suitable for publication in Nature communications. However, there are several points that need to be addressed experimentally.

Major points:

- A weakness of the paper stems from the fact that protein fragments are used. Are cylindrical assemblies also observed when full length SpoVM-mCherry-Cep63 and full length CEP152 are co-overexpressed, not just a CEP152 fragment? Also the in vitro result that full length CEP63 versions can trigger assembly formation is potentially problematic, as the Coomassie gel shows the presence of a CEP63 degradation band. Thus, the observed assemblies might not be caused by full length CEP63 (Figure S2c). There might be reasons (e.g. autoinhibition) why one does not see these assemblies with full length protein on both sides. However, the observed assemblies in vivo recruited the downstream components in a way that looks different from the situation in centrioles (such as PLK4/SAS-6). What happens if SAS-6 assembly is blocked by using a SAS-6 point mutant? Are the recruited SAS-6 amounts / their apparent shape different? Does SpoVM-mCherry-Cep63 FL also recruit endogenous Cep152, SAS6, STIL, PLK4 (currently only shown for exogenously overexpressed centrosomal proteins.)? What happens when SpoVM-mCherry-Cep63 is expressed together with FL CEP152 and the CEP152 interactors CPAP / PLK4 / CEP192? How does this work relate to the role of Cep192 in recruiting Plk4 to the centrosome? The Cep63-Cep152 interaction region appears to overlap with the Cep192-Cep152 interaction region. As Cep192 is present during centriole duplication this is an important question to address.
- The population of elongated molecules in the negative-stain micrograph of Supplementary Figure 3 is small and it is not entirely convincing that the molecules are indeed the Cep63-Cep152 complex. More important for the argument of this manuscript would be to show negative-stain or cryo-EM micrographs of the mCherry-Cep61-P1 + mGFP-Cep152-M4d ring. Poly-lysine coated grids might be of use here to address this.
- How stable are the assemblies observed on slides upon washing with buffer and how does this relate to the FRAP experiments? What is the K_d of the complex in vitro? Furthermore, the high turn-over of the assemblies upon FRAP is striking. Does recovery occur throughout the cylinders or preferentially at ends? If throughout, how does that relate to the proposed assembly model (cooperative, driven by hydrophobic interfaces).
- Complex stoichiometry (Lines 67-75, Lines 159-160 Figure 2a, Figure 3b and supplementary figure 3B). Results suggest a 2:2 stoichiometry for the complex (sedimentation velocity and SEC-MALS experiments – please provide the MW for each construct) and the result was confirmed by the crystal structure. However, what is the MW of the higher order oligomers that was observed at high concentration of the protein fragments. As the presence of this species correlates with the self-assembly of the complex, this is important to address. Are there even high order species observed under higher concentrations – if not, how does that fit to the proposed model? Please provide the runs with the individual proteins as well. Minor point: Please provide the temperature for the SEC-MALS experiment.

- It is not clear to me why the assemblies with a larger diameter are (in absolute terms) longer than those with smaller diameter (Figure 2E). Please explain. One would expect their absolute height to be lower, given that way more subunits have to be added per rise. Is there any correlation between the used concentration of the complex in vitro and the observed diameter of the ring ?
- It has been reported that overexpression of Cep63 or Cep152 cause centriole overduplication. Do any truncated constructs of Cep63 or Cep152, such as Cep63-P1 or Cep152-N70-M4d, stimulate centriole overduplication? Do any Cep63 and Cep152 mutants suppress centriole overduplication when PLK4 is overexpressed ? A prediction of the model would be that they do (dominant negative effect).
- Please change the location of the mCherry / GFP to the other end of the proteins and compare the resulting mCherry / GFP based diameter of the assemblies. According to the model, the assemblies should tolerate this, especially when a flexible linker is used. A plot showing the change of observed diameter in dependence of construct length would also be helpful to appreciate the model further (in the moment scattered throughout the manuscript).

Minor points:

- Many different constructs are used throughout the manuscript that are named in a way that it is hard to read the manuscript. Naming should be changed or a schematic figure provided that shows all constructs together so that readers can read this article smoothly.
- Crystallography: Table S2 – Why has the data been cut at $I/\sigma I$ of 4.39, which is rather (and unusually) high ? It would help to explain how the authors determined the resolution cut-off of the datasets. Rcryst, $I/\sigma I$, CC? Also, authors should add wavelengths used for data collections in the tables. The legend of table 3 says MAD data of the SeMet, but the method says SAD. Also, please expand/clarify the methods section with how exactly the structures were determined. Was an initial model built using the SeMet data and used for molecular replacement of the native P1b5-GGGSE-M4e10 data, followed by usage of that model to solve the structure of P1b5-M4e15?
- Please provide electron density maps of the structure showing at least the key residues that are used in the in vivo studies.
- Figure 3b, did you see any indication of higher order assembly of Cep63(220-541)-Cep152(1140-1295) at higher concentrations ? If not please explain.
- It is hard to understand the details of the interaction between Cep63 and Cep152 only from Figure 3d. It'd be helpful to describe (in the main text or figure) which residues make hydrophobic interactions or polar contacts, to appreciate the nature of the interaction.

• Missing citations:

Sir et al. A primary microcephaly protein complex forms a ring around parental centrioles. *Nature Genetics*, 2011. They already reported the interaction of the C-terminal region of Cep63 with Cep152.

Zhao et al. The Cep63 paralogue Deup1 enables massive de novo centriole biogenesis for vertebrate multiciliogenesis, *Nature Cell Biology*, 2013. The paper is cited in the supplementary material but not in the main text. In this paper the authors have already shown the exact regions for Cep63-Cep152 interaction using similar immunoprecipitation experiments.

Kodani et al. Centriolar satellites assemble centrosomal microcephaly proteins to recruit CDK2 and promote centriole duplication. *eLife*, 2015. The authors proposed a hierarchical assembly of CDK5RAP2, CEP152, WDR62 and CEP63 to the centrosome based on localisation experiments. Since their results do not agree well with the results presented in this paper (see page 6 of Kodani et al, for example), this work should be at least discussed.

- Conservation. CEP63 is not particularly well conserved across metazoan. Therefore, please tone down the reference to strong conservation in the manuscript.

- Page 3 – “requires unusual biochemical and biophysical properties”. To my mind there is nothing particularly unusual about the interaction that is described in this manuscript. I would remove that sentence.
- Figure S1b – a-Flag/a-GFP blots look merged. Please separate the panels or indicate that blots were incubated with two different antibodies.
- Figure S1e,f and other blots. How are the numbers derived? Are they normalized relative to the bait amount pulled down? Blot quantification under conditions of saturated signal is not meaningful and the numbers shown do not always agree with what one sees by eye (e.g. Figure S1i). Please qualify quantification statements.
- There appears to be a discrepancy between the apparent CEP63 self-interaction strength when comparing figure S1g and S1i. Please clarify.
- There also is some discrepancy how strongly the CEP63/CEP152 interaction is affected by the deletion of the hydrophobic motif (not much, Figure S1e,f based on truncation) and how strongly their corresponding self-interaction is affected (strongly, Figure S4a,b). Please expand your explanation on this point.
- As pointed out by the authors, it is difficult to understand why the Alanine mutation of a single hydrophobic motif gives such a strong *in vivo* phenotype. This is not fully backed by the *in vitro* binding data (relatively weak effect). Please qualify the corresponding statements stronger.
- Input signals in different blots are missing. Please provide longer exposures in which the presence of the bands in the inputs shows, if possible.
- Table 3 P4222 should be in the same format as P212121
- Figures 3l,k and the like: Please indicate what the categories 0,1,2,3 are (number of SAS-6 dots?)

Reviewer #2:

Remarks to the Author:

This is an excellent paper that describes fascinating structures formed by Cep63 and Cep152. The authors describe how these two proteins interact through coiled-coil regions and describe how the two proteins are able to assemble into cylindrical structures *in vitro* that are extremely likely to have important significance for centriole structure. These interactions are characterised very thoroughly. The interacting interfaces have been mutated and this prevents recruitment of the complex to centriole *in vivo*. Dramatically the authors show that by targeting full length Cep63 and Cep152 to the membrane surface in U2OS cells, that they can recapitulate formation of the cylindrical structures seen *in vitro*. These structures appear able to recruit Plk4 and in the presence of STIL, Sas6.

In conclusion, this is a carefully executed piece of work that describes the formation of protein complexes likely to be fundamental in directing the assembly of the centriole. I strongly recommend publication of this paper without the need for revision.

Reviewer #3:

Remarks to the Author:

In this manuscript Kim et al. analyze two PCM scaffold proteins, Cep152 and Cep63, that have previously been described to be critical for successful centriole duplication. They found that Cep152 and Cep63 cooperate to recruit downstream interacting proteins as Plk4 by generating a three dimensional architecture through cylindrical self-assembly. The authors further demonstrate that Cep152 and Cep63 interact via a short coiled-coil motif and that hydrophobic residues in both proteins are important for clustering Cep152 and Cep63. My overall reaction to this paper is mixed. Important structural data supporting the authors' conclusions is missing. Moreover, the authors did not show how exactly the cylindrical structure formed by Cep152 and Cep63 leads to recruitment of Plk4 and centriole duplication.

Much more work needs to be performed to develop a coherent and convincing story.

Additional specific points:

- The amount of Cep63 recruited to dimerized Cep152 is enormous. To what extent is Cep63 binding to a Cep152 dimer increased?
- The authors used FRAP analysis to analyze the self-assembly in Fig. 2i. It is necessary to show earlier time points, between 1.7 min and 5.0 minutes, to draw a clear conclusion.
- The confocal images in Fig. 1e needs quantification. Is the $\Delta M4e$ fragment ($\Delta 68$ dimer of Cep152) predominantly localized to only one of the two centrosomes? Figure 1e would suggest this.
- To demonstrate the nature of Cep152 and Cep63 self-assembly, the authors used X-ray crystallography. What resolution do the authors present here? In Figs. 3 c and d they need to show high resolution structures of dimerization and interactions.
- at what time point during G1 does the self-assembly of Cep63 and Cep152 occur? Does this process already start right after mitosis is completed? It is important to analyze when during G1 Plk4 is recruited by Cep63 and Cep152.
- The authors analyze recruitment of Sas-6 to the centrosome dependent on the 8A (Cep63) and 6A (Cep152) mutant. I would have expected that they determine the level of Plk4 at the centrosome as Plk4 is directly recruited by Cep152. These data need to be included. Is Plk4 lacking Cep152 binding sites as a control not recruited to the centrioles?
- As both Cep63 and Cep152 are microcephaly proteins, it is interesting to see whether mutations found in MCPH patients play a role in self-assembly or recruitment of Plk4 and Sas6. Did the authors investigate specific MCPH mutations? Impairment of Cep63 binding to Cep152 or failure to self-assemble would be a nice confirmation of their model.
- what is the phenotype of the Cep83 8A and Cep152 6A mutants? Does their expression in cells impair centriole duplication?

other points:

- Fig. 1a right WB, Flag input signals are too strong making it difficult to judge equal loading of the samples. In Figs. 1 b and c, Flag inputs are missing. The figure legend of Fig. 1a is too short and not informative.
- the terminology the authors used for the Cep152 or Cep63 mutants is very confusing, for example, what does Cep152 M4 and M6 mean?
- For the sedimentation velocity analysis, purified Cep63 and Cep152 fragments were used. The authors need to show details about grade (%) of purity of the fragments
- Figs. 3 i and k, 4 d and f: I guess that colored squares denote the number of centrioles. This needs to be included into the figure legend.

Responses to the reviewers' comments:

In response to the reviewers' comments, we provide the following new data to further strengthen our original manuscript. Point-by-point responses are shown further below.

1. New Fig. 2g – Summary of diameter measurements for the three types of cylindrical assemblies, including the assemblies made of the Cep63-mCherry (C-terminally tagged mCherry) and mGFP-Cep152. The results are consistent with the proposed model in Fig. 5c
2. New Supplementary Fig. 1k – Quantified data for Fig. 1e.
3. New Supplementary Fig. 1l – Coimmunoprecipitation data demonstrating the strength of the Cep63-Cep152 interaction in comparison to a weak Cep192-Cep152 interaction.
4. New Coomassie-stained gels (Supplementary Fig. 2a and Supplementary Fig. 3a, 3c) showing the purity of protein samples.
5. New Supplementary Fig. 2c – Stability test for the self-assemblies in Fig. 2c to demonstrate that these assemblies are largely stable under a physiological buffer condition. To reconcile this observation with the dynamic exchange of components between the assembly and its surroundings shown in Fig. 2j, an explanation in a thermodynamic view is provided in line 162.
6. New Supplementary Fig. 2d – Quantified data assessing the effect of different concentrations of the Cep63 P1-Cep152 M4d complex on the morphology of the cylindrical self-assembly.
7. New Supplementary Fig. 2e – Examination of the effect of Cep63-mCherry on the diameter of the self-assembly (the data are incorporated to generate the new Fig. 2g).
8. New Supplementary Fig. 3a – An improved negative-stained TEM image used for replacement.
9. New Supplementary Fig. 3b – Sedimentation velocity data for a second complex (a longer form described in the original manuscript).
10. New Supplementary Fig. 3d – SEC-MALS analyses with serially diluted samples used to determine K_d values for a Cep63-Cep152 complex.
11. New Supplementary Fig. 3e, 3f – Electron density maps for critical residues and summary of important interactions in forming the four-helix bundle.
12. New Supplementary Fig. 7h – *In vivo* self-assemblies generated with the full-length Cep63 and Cep152.
13. New Supplementary Fig. 8c – Representative images showing the effect of the oligomerization-defective Sas6 (F131E) mutation.
14. New Supplementary Fig. 8d – Data showing a dominant-negative effect of truncated Cep63 P1 or Cep152 M4d.
15. New Supplementary Fig. 9 – Schematic diagram illustrating all the constructs used in this study.
16. Longer or shorter exposed immunoblots to replace the original blots in Fig. 1a–c, and Supplementary Fig. 1j.

Outlined below are our point-by-point answers to the reviewers' comments.

Reviewer #1 (Remarks to the Author):

Kim et al. describe that in vitro and in vivo a complex of CEP63 and CEP152 fragments is able to cooperatively form cylindrical assemblies that resemble the observed CEP63 and CEP152 localisation around centrioles in cells. The authors complement these findings by providing a high-resolution structural description of a critical protein-protein interaction interface involved in this process and show that this interface is essential for centriole duplication. Their structural work also suggests models for the exact organisation of CEP63 and CEP152 assemblies around centrioles that is overall in agreement with previous localisation studies with both proteins in cells.

This work is potentially an important contribution towards understanding how the assembly of such a large and complex protein assembly as the centriole can arise from the scaffold forming / self-assembly properties of a few of its core constituents. As such this work would be suitable for publication in Nature communications. However, there are several points that need to be addressed experimentally.

Major points:

- A weakness of the paper stems from the fact that protein fragments are used. Are cylindrical assemblies also observed when full length SpoVM-mCherry-Cep63 and full length CEP152 are co-overexpressed, not just a CEP152 fragment? Also the in vitro result that full length CEP63 versions can trigger assembly formation is potentially problematic, as the Coomassie gel shows the presence of a CEP63 degradation band. Thus, the observed assemblies might not be caused by full length CEP63 (Figure S2c). There might be reasons (e.g. autoinhibition) why one does not see these assemblies with full length protein on both sides.*

Answer: Coexpression of SpoVM-mCherry-63 and mGFP-Cep152 in U2OS cells also yielded cylindrical assemblies morphologically indistinguishable from those formed with mCherry-Cep63 P1 and GFP-Cep152 M4d. Consistent with other data provided in the manuscript, mGFP-Cep152 signals were detected at the outskirts of the Cep63 signals (**provided as a new Supplementary Fig. 7h**). One notable observation is that the efficiency of generating the assemblies with the full-length forms was a few-fold lower than that with the shorter Cep63 P1 and Cep152 M4d.

The reviewer questions whether there could be an autoinhibition with the full-length form and also concerns about the degraded fragments contributing to the self-assembly. While these are valid concerns, we were not able to purify these long coiled-coil proteins without any degradation (coiled-coil proteins are considered very difficult to purify). Although we cannot eliminate a potential autoinhibition, we currently think that the reduction in the assembling efficiency is likely due to the strong non-specific aggregative nature of these coiled-coil proteins. Coiled-coil proteins are known to be highly vulnerable to nonspecific local interactions due to their characteristic flat free energy landscape (*Lupas AN, Trends Biochem. Sci, 42:130. 2017*). This point is mentioned in line 143. Whether an autoinhibitory mechanism exists in the process of self-assembly and whether it is post-translationally controlled would be important future questions that need further investigation.

However, the observed assemblies in vivo recruited the downstream components in a way that looks different from the situation in centrioles (such as PLK4/SAS-6). What happens if SAS-6 assembly is blocked by using a SAS-6 point mutant? Are the recruited SAS-6 amounts / their apparent shape different?

Answer: It is well documented that Plk4-dependent STIL binding and subsequent Sas6 phosphorylation are critical for proper Sas6 recruitment and procentriole assembly (Ohta M, et al, *Nat Commun.* 5:5267. 2014; Dzhinzhev NS, et al, *Curr. Biol.* 24:2526. 2014). Although Plk4 and Sas6 recruited to the Cep63-Cep152 assembly did not exhibit a distinct dot-like morphology as observed with endogenous proteins, we observed that treatment of cells with a phosphatase 2A inhibitor was necessary to detect clustered Plk4 and Sas6 signals around an assembly (as described in line 305). Thus, we speculate that failure to fully activate Plk4 under overexpressed conditions contributed to the morphologically less-focused Plk4 and Sas6 signal.

As suggested, we examined the pattern of Sas6 WT or its oligomerization-defective Sas6 (F131E) mutant (Keller D, et al, *JCB*:204:97. 2014) recruited to SpoVM-mCherry-Cep63•mGFP-Cep152 M6 assemblies. Notably, while WT was detected as a congregated dot-like morphology, Sas6 (F131E) was observed only as weak and unfocused signals (**provided as a new Supplementary Fig. 8c**). This finding concurs with the Keller D, et al's observation. This finding suggests that the Sas6 (F131E) mutant can be recruited to the self-assembly but cannot maintain its focused localization because of its inability to form an organized cartwheel structure.

Does SpoVM-mCherry-Cep63 FL also recruit endogenous Cep152, SAS6, STIL, PLK4 (currently only shown for exogenously overexpressed centrosomal proteins.)?

Answer: We found that SpoVM-mCherry-Cep63 FL alone can recruit endogenous Cep152, Plk4, and Sas6 at an ectopic site. However, these recruited endogenous proteins were not sufficient to cooperate with SpoVM-mCherry-Cep63 FL to generate an assembly. We speculate that these recruited proteins may not have exceeded their threshold concentrations to drive the formation of the self-assembly. This result is provided at the end of this letter below for reviewing purposes.

What happens when SpoVM-mCherry-Cep63 is expressed together with FL CEP152 and the CEP152 interactors CPAP / PLK4 / CEP192? How does this work relate to the role of Cep192 in recruiting Plk4 to the centrosome? The Cep63-Cep152 interaction region appears to overlap with the Cep192-Cep152 interaction region. As Cep192 is present during centriole duplication this is an important question to address.

Answer: As the reviewer pointed out, it has been reported that Cep192 binds to Cep152. However, the level of the Cep192-Cep152 interaction is very low in comparison to that of the Cep63-Cep152 interaction (**provided as a new Supplementary Fig. 11**). Consistent with this finding, Cep192 localizes significantly (~40 nm) inner side of the Cep152 toroid (Lawo S, et al., *NCB* 14:1148. 2012; Park SY, et al., *NSMB* 21:696. 2014). In addition, a chemical cross-linking approach identified Cep63, Cep152, and Cep57 as a centrosomal complex, but not Cep192 (Lukinavicius G, et al, *Curr Biol.* 23:265. 2013). Thus, while we cannot eliminate the physiological significance of the Cep192-Cep152 interaction, we think that this interaction may not be strong enough to alter the Cep63-Cep152 interaction. This finding is stated in line 60 and Supplementary Fig. 1 legend.

• The population of elongated molecules in the negative-stain micrograph of Supplementary Figure 3 is small and it is not entirely convincing that the molecules are indeed the Cep63-Cep152 complex. More important for the argument of this manuscript would be to show negative-stain or cryo-EM

micrographs of the mCherry-Cep61-P1 + mGFP-Cep152-M4d ring. Poly-lysine coated grids might be of use here to address this.

Answer: A much improved image is now provided along with the protein used for the staining (**new Supplementary Fig. 3a**). Measurement of the length of the rod-shaped complex yielded a contour length of 31 ± 4.11 nm ($n = 30$) (stated in line 178). Although a cryo-EM approach would be much desired to understand the architecture of the Cep63 P1•Cep152 M4d assembly, a new level of effort will be required to achieve this goal. It seems to be beyond of the scope of this manuscript.

• How stable are the assemblies observed on slides upon washing with buffer and how does this relate to the FRAP experiments?

Answer: We found that the Cep63 P1•Cep152 M4d assemblies are largely stable for up to five days. Given that self-assembly is a spontaneous free energy-minimization process and reversing this process would require overcoming the free energy barrier, this is somewhat expected. As the reviewer pointed out, a FRAP experiment in Fig. 2j showed that the self-assembly is yet very dynamic, exchanging its components with those in the surroundings. This could be possible if the binding energy of incoming molecules (which is a favorable step in self-assembly), generated through the spontaneous assembly process, is harnessed to induce the unfavorable dissociation process for the molecules present in the assembly. This view is discussed in line 162.

What is the K_d of the complex in vitro?

Answer: We were not able to purify individual Cep63 P1 and Cep152 M4d (they are insoluble unless forming a complex), and therefore could not carry out isothermal titration calorimetry analysis. As an alternative, we purified the Cep63 P1b5•Cep152 M4e15 complex (shown in the Fig. 3c crystal structure) and carried out serial SEC-MALS analyses to estimate the K_d values for forming the heterotetrameric complex. As shown in the **new Supplementary Fig. 3d**, we found that dissociation of the Cep63 P1b5•Cep152 M4e15 complex apparently occurs in two distinguishable steps. The crystal structure in Fig. 3c suggests that the M4e15-M4e15 backbone interaction is much stronger than the two P1b5 molecules binding to the M4e15 dimer. Thus, we propose that the stronger interaction ($K_d \sim 60$ μ M) represents the M4e15-M4e15 interaction, while the weaker interaction ($K_d \sim 117$ μ M) belongs to the two P1b5 molecules binding to the M4e15 dimer. This is stated in line 194 and the Supplementary Fig. 3 legend.

Furthermore, the high turn-over of the assemblies upon FRAP is striking. Does recovery occur throughout the cylinders or preferentially at ends? If throughout, how does that relate to the proposed assembly model (cooperative, driven by hydrophobic interfaces).

Answer: Although the interpretation of SIM-TIRF data could be somewhat limited, because TIRF preferentially illuminates specimen within ~ 100 nm, we think the exchange may likely occur throughout the cylindrical assembly. To address the reviewer's question, FRAP on the total assembly combined with super-resolution SIM could be a good approach. However, because of background fluorescence signals coming from free (unassembled) proteins present in solution, we could not detect the cylindrical assemblies without restricting the imaging to a limited specimen region using TIRF. Based on the negative exponential distribution pattern of the self-assemblies shown in Fig. 2d and the **new Supplementary Fig. 2d**, we propose that the assembly process occurs in a continuous-time stochastic fashion, presumably accumulating incoming molecules at the top end of the cylindrical assembly (as proposed in Fig. 5c). However, the FRAP recovery observed between the assembly and its surroundings in Fig. 2j suggests that the assembly is likely flexible through the mechanism discussed above. Therefore, we added this point in the Fig. 5 legend.

The cooperativity comes from the level of forming the heterotetrameric building block, as demonstrated in Fig. 1b, 1c and Supplementary Fig. 1i and 1j, and this could be fueled by the hydrophobic clustering effect as proposed in Fig. 4c. It should be worth noting that a well characterized actin assembly also shows a similar negative-exponential-distribution pattern (*Burlacu S, et al, Am. J. Physiol. 262:C569. 1992; Sept D, et al, Biophys. J. 77:2911. 1999*). Remarkably, actin forms a one-dimensional polymer, whereas the Cep63 P1•Cep152 M4 complex forms a three-dimensional higher-order structure, thus adding an additional layer of complexity in the mechanism of building this higher-order architecture.

• Complex stoichiometry (Lines 67-75, Lines 159-160 Figure 2a, Figure 3b and supplementary figure 3B). Results suggest a 2:2 stoichiometry for the complex (sedimentation velocity and SEC-MALS experiments – please provide the MW for each construct) and the result was confirmed by the crystal structure.

Answer: As requested, expected MW of the three constructs used in Fig. 2a, 3b, **new Supplementary Fig. 3b**, and Supplementary Fig. 3c are provided in line 71, line 181, and line 193 (also Supplementary Fig. 3 legend), respectively.

However, what is the MW of the higher order oligomers that was observed at high concentration of the protein fragments. As the presence of this species correlates with the self-assembly of the complex, this is important to address. Are there even high order species observed under higher concentrations – if not, how does that fit to the proposed model? Please provide the runs with the individual proteins as well. Minor point: Please provide the temperature for the SEC-MALS experiment.

Answer: Since the fast-sedimenting species shown in Fig. 2a shows ~4 S, we suspect that this species could be an octameric form. This speculation is based on scaling laws that a dimer is expected to sediment approximately 1.5 times as fast as the monomer (*Brown PH, et al, Biophys. J. 90:4651. 2006*). As the reviewer pointed out, purifying this higher-order complex and determining its MW would be very informative. Attempts to purify this species have been failed because of sample dilution that occurs during SEC-MALS. We suspect that this higher-order complex is transiently formed during sedimentation velocity ultracentrifugation (likely driven from equilibrium), and it can be dissociated back to the tetrameric form upon dilution. This is highly likely, considering that the tetrameric form can be further dissociated into individual fragments as shown in the **new Supplementary Fig. 3d** (described above). We have not observed any species faster than the proposed octameric form under our experimental conditions. We hypothesize that the Cep63 P1•Cep152 M4d heterotetramer is the primary building block for the cylindrical assembly and the higher-order complex is likely the result of stochastic interactions among the building block molecules. A similar higher-order complex was also observed for a longer Cep63 (220–541)•Cep152 (1140–1295) (**new Supplementary Fig. 3b**) and this finding is described in line 182.

As stated above, we were not able to purify individual Cep63 P1 or Cep152 M4d, because they were insoluble, unless they form a complex with each other. Temperature for the SEC-MALS (4°C) is now provided in the Methods.

• It is not clear to me why the assemblies with a larger diameter are (in absolute terms) longer than those with smaller diameter (Figure 2E). Please explain. One would expect their absolute height to be lower, given that way more subunits have to be added per rise.

Answer: We agree with the reviewer that this phenomenon was somewhat unexpected, providing that the local concentration of the building block in solution should be similar. Notably, all the data (Fig.

2c, **new Supplementary Fig. 2d**, Supplementary Fig. 2g, and **new Supplementary Fig. 2i**) equally show that the self-assemblies with larger diameters are taller. Our data provided in **new Supplementary Fig. 3d** show somewhat moderate affinities for forming the Cep63 P1•Cep152 M4 complex (60 μM and 117 μM , see above), suggesting that the K_d for inter-building block interactions could be significantly weaker than that for forming the complex. Under these circumstances, the free form of the complex in solution is likely in excess. Therefore, the assembly process may not be limited by available building blocks in the surroundings and the height of the assemblies could be proportional to the stability of the assembled architecture. This point is stated in line 117.

Is there any correlation between the used concentration of the complex in vitro and the observed diameter of the ring?

Answer: This is a very important question, but difficult to accurately address it. It has been reported that there are ~ 540 Cep152 molecules per centrosome (Bauer M, *EMBO J.* 35:2152. 2016). Assuming that the area of Cep152-loaded pericentriolar region is $\sim 0.4 \mu\text{m}^2$ [$0.4 \mu\text{m}$ (D) $\times \pi \times 0.3 \mu\text{m}$ (H)] (Cep152 localizes to the proximal half region of a centriole, which is typically $0.6 \mu\text{m}$ tall), this area is $\sim 2 \times 10^8$ times smaller than the coverslip area occupied by the $10 \mu\text{l}$ of the sample. With $10 \mu\text{l}$ of $6 \mu\text{M}$ complex loaded, there will be 1.8×10^5 molecules/ $0.4 \mu\text{m}^2$, if all the molecules in solution are recruited to the surface of the coverslip. Assuming that $\sim 10\%$ of the total molecules are being recruited to the surface, we estimate that the concentration that we are using could be ~ 30 times higher than that of endogenous Cep152 found around a centriole.

• It has been reported that overexpression of Cep63 or Cep152 cause centriole overduplication. Do any truncated constructs of Cep63 or Cep152, such as Cep63-P1 or Cep152-N70-M4d, stimulate centriole overduplication? Do any Cep63 and Cep152 mutants suppress centriole overduplication when PLK4 is overexpressed? A prediction of the model would be that they do (dominant negative effect).

Answer: We examined whether the expression of truncated Cep63 P1 or Cep152-N70-M4d fragment alters centriole duplication by determining the number of Sas6 recruited to centrosomes. As the reviewer anticipated, we found that both of these constructs induced a dominant-negative effect, crippling the recruitment of Sas6 under both without and with Plk4 coexpression. This finding is provide as the **new Supplementary Fig. 8d**.

• Please change the location of the mCherry / GFP to the other end of the proteins and compare the resulting mCherry / GFP based diameter of the assemblies. According to the model, the assemblies should tolerate this, especially when a flexible linker is used.

Answer: As suggested, we generated self-assemblies using Cep63 C-terminally tagged with mCherry. The result is provided in the **new Supplementary Fig. 2i**. This finding further strengthens our originally proposed model that the Cep63•Cep152 complex is radially arranged from the axis of the cylindrical assembly. This new finding is described in line 133.

A plot showing the change of observed diameter in dependence of construct length would also be helpful to appreciate the model further (in the moment scattered throughout the manuscript).

Answer: As recommended, we now provide a summary plot in the **new Fig. 2g**, showing the variations in assembly dimeters with different constructs. Thank you for the suggestion.

Minor points:

• *Many different constructs are used throughout the manuscript that are named in a way that it is hard to read the manuscript. Naming should be changed or a schematic figure provided that shows all constructs together so that readers can read this article smoothly.*

Answer: As suggested, we now provide a **new Supplementary Fig. 9**, summarizing all the constructs used in this study.

• *Crystallography: Table S2 – Why has the data been cut at $I/\sigma I$ of 4.39, which is rather (and unusually) high? It would help to explain how the authors determined the resolution cut-off of the datasets. Rcryst, $I/\sigma I$, CC? Also, authors should add wavelengths used for data collections in the tables. The legend of table 3 says MAD data of the SeMet, but the method says SAD. Also, please expand/clarify the methods section with how exactly the structures were determined. Was an initial model built using the SeMet data and used for molecular replacement of the native P1b5-GGGSE-M4e10 data, followed by usage of that model to solve the structure of P1b5-M4e15?*

Answer: We cut the data at $I/\sigma I$ of 4.39 because we wanted to keep the completeness at least $> 70\%$ level. As requested, the wavelength is now provided into the Supplementary Table 2 and 3. Yes, SAD is correct. Now, we changed it in the Supplementary Table 3.

As suggested, we expanded the Structure determination section (in Methods) and provided detailed methods that we used to determine the structures.

• *Please provide electron density maps of the structure showing at least the key residues that are used in the in vivo studies.*

Answer: As suggested, we now provided a **new Supplementary Fig. 3e**, showing the electron densities for the residues that we mutated.

• *Figure 3b, did you see any indication of higher order assembly of Cep63(220-541)-Cep152(1140-1295) at higher concentrations? If not please explain.*

Answer: We carried out sedimentation velocity analysis for the Cep63(220-541)•Cep152(1140-1295) complex and showed that this longer form also forms a faster-sedimenting species, matching to the MW of a predicted octameric complex, a phenomena similar to that with the shorter Cep63 P1•Cep152 M4d complex shown in Fig. 2a. This finding is stated in line 181.

• *It is hard to understand the details of the interaction between Cep63 and Cep152 only from Figure 3d. It'd be helpful to describe (in the main text or figure) which residues make hydrophobic interactions or polar contacts, to appreciate the nature of the interaction.*

Answer: As requested, we now provided a summary of all the important interactions in a **new Supplementary Fig. 3f**.

• *Missing citations:*

Sir et al. A primary microcephaly protein complex forms a ring around parental centrioles. Nature Genetics, 2011. They already reported the interaction of the C-terminal region of Cep63 with Cep152.

Zhao et al. The Cep63 paralogue Deup1 enables massive de novo centriole biogenesis for vertebrate multiciliogenesis, Nature Cell Biology, 2013. The paper is cited in the supplementary material but not in the main text. In this paper the authors have already shown the exact regions for Cep63-Cep152 interaction using similar immunoprecipitation experiments.

Kodani et al. Centriolar satellites assemble centrosomal microcephaly proteins to recruit CDK2 and promote centriole duplication. eLife, 2015. The authors proposed a hierarchical assembly of CDK5RAP2, CEP152, WDR62 and CEP63 to the centrosome based on localisation experiments. Since their results do not agree well with the results presented in this paper (see page 6 of Kodani et al, for example), this work should be at least discussed.

Answer: This was our oversight! Now, we provided the Sir et al and the Zhao papers in line 44. Also, we discussed the Kodani paper in line 232.

• *Conservation. CEP63 is not particularly well conserved across metazoan. Therefore, please tone down the reference to strong conservation in the manuscript.*

Answer: As suggested, we have incorporated this point in line 205.

• *Page 3 – “requires unusual biochemical and biophysical properties”. To my mind there is nothing particularly unusual about the interaction that is described in this manuscript. I would remove that sentence.*

Answer: As far as we are aware, there is no report describing intracellular proteins that cooperatively self-assemble into a three-dimensional cylindrical architecture (note that actin filaments and microtubules are one-dimensional assemblies). In our view, it is appropriate to leave the statement as it is because of this unusual capacity of Cep63 P1 and Cep152 M4d fragments in forming an unprecedented helix-based cylindrical self-assembly. This self-assembly is distinct from all the previously characterized helix-based oligomeric structures summarized in a recent review (*Lupas, AN, Trends Biochem. Sci, 42:130. 2017*).

• *Figure S1b – a-Flag/a-GFP blots look merged. Please separate the panels or indicate that blots were incubated with two different antibodies.*

Answer: Now, this has been taken care of.

• *Figure S1e,f and other blots. How are the numbers derived? Are they normalized relative to the bait amount pulled down? Blot quantification under conditions of saturated signal is not meaningful and the numbers shown do not always agree with what one sees by eye (e.g. Figure S1i). Please qualify quantification statements.*

Answer: As suggested, we re-quantified the signal intensities in the blot and normalized them to the amounts of immunoprecipitated ligands. This is stated in the Supplementary Fig. 1 legend. The new intensity value is provided in red. This change does not alter the conclusion of the binding assay. Thank you for the comment.

• *There appears to be a discrepancy between the apparent CEP63 self-interaction strength when comparing figure S1g and S1i. Please clarify.*

Answer: We admit that there are variations in the coimmunoprecipitation efficiency. They are most likely due to the variations in the amounts of ligands immunoprecipitated. Since the ligand blots (bottom blots) were generated after blotting input and co-precipitated target proteins, we cannot directly compare the ligand and co-precipitated target amount from one blot with those from another blot. It is however important to make sure that the precipitated ligand amounts are similar among different lanes to allow meaningful comparisons among coprecipitated targets.

• *There also is some discrepancy how strongly the CEP63/CEP152 interaction is affected by the*

deletion of the hydrophobic motif (not much, Figure S1e,f based on truncation) and how strongly their corresponding self-interaction is affected (strongly, Figure S4a,b). Please expand your explanation on this point.

Answer: The interactions shown in Supplementary Fig. 1e and 1f are between Cep63 and Cep152, whereas those shown in Supplementary Fig. 4a and 4b are between homomeric Cep63 P1 or homomeric Cep152 M4e. These observations suggest that the hydrophobic motif is more important for the heteromeric Cep63-Cep152 interaction. In the presence of both Cep63 P1 and Cep152 M4d, these hydrophobic motifs become less important because of the presence of the strong heterotetrameric interactions (the region shown in the crystal structure in Fig. 3d). This is clearly stated in line 246.

• As pointed out by the authors, it is difficult to understand why the Alanine mutation of a single hydrophobic motif gives such a strong in vivo phenotype. This is not fully backed by the in vitro binding data (relatively weak effect). Please qualify the corresponding statements stronger.

Answer: This is a good point. It is at all times difficult to know whether the biochemical activity can be proportionally translatable to the biological activity. Our coimmunoprecipitation assay examines only the interaction ability between soluble proteins under physiological buffer conditions but does little to understand whether and, if so, how this interaction contribute to the formation of a three-dimensional functional assembly. This apparent discrepancy allows us to speculate that, in addition to mediating the Cep63 P1 and Cep152 M4d interaction, the hydrophobic motif could architecturally contribute to the formation of the self-assembly. This point is stated in line 260.

• Input signals in different blots are missing. Please provide longer exposures in which the presence of the bands in the inputs shows, if possible.

Answer: As suggested, longer exposed gels are provided for Fig. 1b, 1c, and Supplementary Fig. 2j.

• Table 3 P4222 should be in the same format as P212121

Answer: We made corrections, as suggested. Thank you.

• Figures 3I,k and the like: Please indicate what the categories 0,1,2,3 are (number of SAS-6 dots ?)

Answer: We now made it clear that the numbers indicate Sas6 dot signals per cell. This is stated in the Fig. 3 legend. Thank you for many great comments!

Reviewer #2 (Remarks to the Author):

This is an excellent paper that describes fascinating structures formed by Cep63 and Cep152. The authors describe how these two proteins interact through coiled-coil regions and describe how the two proteins are able to assemble into cylindrical structures in vitro that are extremely likely to have important significance for centriole structure. These interactions are characterised very thoroughly. The interacting interfaces have been mutated and this prevents recruitment of the complex to centriole in vivo. Dramatically the authors show that by targeting full length Cep63 and Cep152 to the membrane surface in U2OS cells, that they can recapitulate formation of the cylindrical structures seen in vitro. These structures appear able to recruit Plk4 and in the presence of STIL, Sas6.

In conclusion, this is a carefully executed piece of work that describes the formation of protein complexes likely to be fundamental in directing the assembly of the centriole. I strongly recommend publication of this paper without the need for revision.

Answer: We appreciate this comment! Thank you for recognizing the importance of this work!

Reviewer #3 (Remarks to the Author):

In this manuscript Kim et al. analyze two PCM scaffold proteins, Cep152 and Cep63, that have previously been described to be critical for successful centriole duplication. They found that Cep152 and Cep63 cooperate to recruit downstream interacting proteins as Plk4 by generating a three dimensional architecture through cylindrical self-assembly. The authors further demonstrate that Cep152 and Cep63 interact via a short coiled-coil motif and that hydrophobic residues in both proteins are important for clustering Cep152 and Cep63. My overall reaction to this paper is mixed. Important structural data supporting the authors' conclusions is missing. Moreover, the authors did not show how exactly the cylindrical structure formed by Cep152 and Cep63 leads to recruitment of Plk4 and centriole duplication. Much more work needs to be performed to develop a coherent and convincing story.

Answer: Now, we provide detailed structural data, showing how Cep63 P1b5 and Cep152 M4e15 interact with each other to form a heterotetrameric complex. New figures are provided as

Supplementary Fig. 3e and 3f.

We and others have published how Cep152 recruits Plk4 and promotes centriole duplication (Kim TS, et al, PNAS 110:E4849. 2013; Sonnen KF, et al., JCS 126:3223. 2013). Furthermore, we published the cocrystal structure of the Plk4 cryptic polo-box domain (CPB) and the Cep152 N-terminal region (the fragment marked as N70 in this manuscript) (Park SY, et al, NSMB 21:696. 2014). These are described in line 270. Various other studies have demonstrated how Plk4 recruits STIL and Sas6 (Ohta M, et al, Nat Commun. 5:5267. 2014; Dzhinzhev NS, et al., Curr. Biol. 24:2526. 2014; Kratz AS, et al., Biol. Open 4:370. 2015), and how recruited Sas6 forms the cartwheel structure of a centriole (Guichard P, Bioassay 40:e1700241. 2018, review). This is described in line 301. Sorry that we could not fully introduce these works due to limited space.

Additional specific points:

- The amount of Cep63 recruited to dimerized Cep152 is enormous. To what extent is Cep63 binding to a Cep152 dimer increased? (The reviewer specified: "I am referring to Fig. 1b,c and Fig. 3g and their significance.")

Answer: We quantified the level of immunoblot signal intensities by using Image J, and found that the Cep152-Cep152 interaction increased ~7-9 folds in the presence of Cep63 (Fig. 1b and 1c). Strikingly, this cooperative interaction was almost completely abrogated when the Cep152-Cep152 interaction is disrupted by the 6A mutations (Fig. 3g). Consistent with these findings, Cep63 and Cep152 were mutually required to localize to centrosomes (Fig. 3h and 3j) and their centrosome-localization capacity was severely compromised by the 6A mutation (Fig. 3j). Likewise, the 6A mutant was largely defective in recruiting downstream Sas6 to the procentriole assembly site (Fig. 3k). These observations strongly support the argument that the formation of the Cep63-Cep152 complex is critical for not only their centrosomal localization but also their function in promoting centriole duplication.

- The authors used FRAP analysis to analyze the self-assembly in Fig. 2i. It is necessary to show earlier time points, between 1.7 min and 5.0 minutes, to draw a clear conclusion.

Answer: A representative image for 3.3 min time point is provided. The provided time-lapse video (Supplementary Movie 3) will allow continuous viewing of all processes. Video starts 30 minutes

after placing the mCherry-Cep63 P1•mGFP-Cep152 M4d complex on a coverslip for SIM-TIRF at RT.

- The confocal images in Fig. 1e needs quantification. Is the $\Delta M4e$ fragment ($\Delta 68$ dimer of Cep152) predominantly localized to only one of the two centrosomes? Figure 1e would suggest this.

Answer: As requested, quantified data for Fig. 1e are provided as a **new Supplementary Fig. 1k**. Quantification was carried out with three independent experiments (100 to 208 cells/sample/experiment) and the resulting data were statically analyzed.

- To demonstrate the nature of Cep152 and Cep63 self-assembly, the authors used X-ray crystallography. What resolution do the authors present here? In Figs. 3 c and d they need to show high resolution structures of dimerization and interactions.

Answer: Both crystal structures were solved at 2.5 Å. All the data collection and refinement statistics are provided in Supplementary Table 2 and 3.

As requested, now we provide **new Supplementary Figure 3e and 3f**, showing electron densities for important residues and all the interactions observed in the formation of the heterotetrameric complex.

- at what time point during G1 does the self-assembly of Cep63 and Cep152 occur? Does this process already start right after mitosis is completed? It is important to analyze when during G1 Plk4 is recruited by Cep63 and Cep152.

Answer: Cep63 and Cep152 are recruited to daughter centrioles during the late stage of the G1 phase. Therefore, the assembly of the Cep63•Cep152 complex presumably occurs immediately after their recruitment to centrosomes. It is well documented that Cep152 (also together with Cep192) plays a key role in recruiting Plk4 to pericentriolar region (Kim TS, et al, PNAS 110:E4849. 2013; Sonnen KF, et al, JCS 126:3223. 2013). All the quantifications were carried out among interphase cells where Cep63, Cep152, Plk4, and Sas6 are present.

- The authors analyze recruitment of Sas-6 to the centrosome dependent on the 8A (Cep63) and 6A (Cep152) mutant. I would have expected that they determine the level of Plk4 at the centrosome as Plk4 is directly recruited by Cep152. These data need to be included. Is Plk4 lacking Cep152 binding sites as a control not recruited to the centrioles?

Answer: As suggested, we quantified the level of Plk4 recruitment in cells expressing either the Cep63 (8A) or Cep152 (6A) mutant. The results are provided at the end of the letter for reviewing purposes. The reason that we did not incorporate the data into the manuscript is that Plk4 recruitment is regulated by not only Cep152 but also another inner scaffold Cep192, thus essentially not allowing to properly interpret resulting data. For instance, it has been reported that depletion of Cep152 rather increases the centrosome-recruited Plk4 2-3 folds (Kim TS, et al, PNAS 110:E4849. 2013; Sonnen KF, et al, JCS 126:3223. 2013), even though it is well documented that the timing of Cep152 recruitment controls Plk4-dependent centriole duplication (Novak, ZA, et al., Curr. Biol. 24:1276. 2014).

The control lacking the Plk4-binding motif is provided in the Supplementary Fig. 6. As expected, Plk4 was not recruited to the Cep63•Cep152 self-assemblies.

- As both Cep63 and Cep152 are microcephaly proteins, it is interesting to see whether mutations found in MCPH patients play a role in self-assembly or recruitment of Plk4 and Sas6. Did the

authors investigate specific MCPH mutations? Impairment of Cep63 binding to Cep152 or failure to self-assemble would be a nice confirmation of their model.

Answer: This is a very important suggestion. However, a search of literatures did not yet yield any MCPH mutations within the Cep63 P1 and Cep152 M4d regions, although multiple mutations outside of these fragments were found. Considering the importance of this research direction, we will continue to look for pathogenic mutations within these regions and investigate the significance of these mutations in altering the structure and function of Cep63 P1 and Cep152 M4d-driven self-assembly.

- what is the phenotype of the Cep83 8A and Cep152 6A mutants? Does their expression in cells impair centriole duplication?

Answer: We examined the effect of Cep63 (8A) and Cep152 (6A) mutations in U2OS cells and found that these mutants were severely impaired in recruiting Sas6, a key structural component of the cartwheel structure of a centriole (Guichard P, *Bioassay* 40:e1700241. 2018, review). The data are presented in Fig. 3i and 3k.

other points:

- Fig. 1a right WB, Flag input signals are too strong making it difficult to judge equal loading of the samples. In Figs. 1 b and c, Flag inputs are missing. The figure legend of Fig. 1a is too short and not informative.

Answer: As suggested, a weakly exposure gel is now provided for Fig. 1b. In addition, longer exposed gels are provided for Fig. 1b, 1c to show inputs.

- the terminology the authors used for the Cep152 or Cep63 mutants is very confusing, for example, what does Cep152 M4 and M6 mean?

Answer: To help avoid any confusion, we now provide a **new Supplementary Fig. 9**, summarizing all the constructs used in this study.

- For the sedimentation velocity analysis, purified Cep63 and Cep152 fragments were used. The authors need to show details about grade (%) of purity of the fragments

Answer: As recommended, Coomassie-stained gels are provided to show the purity of the proteins used for sedimentation velocity analyses in Supplementary Fig. 2a and Supplementary Fig. 3a, 3c.

Figs. 3 i and k, 4 d and f: I guess that colored squares denote the number of centrioles. This needs to be included into the figure legend.

Answer: Now, we made it clear that the numbers indicate Sas6 dot signals per cell. This is stated in the Fig. 3 legend. Thank you for several great comments!

Sincerely,

Kyung Lee

A figure and four graphs are attached below (for reviewing purposes)

For reviewing purposes only:

Reviewer #1's comment #3:

Expression of SpoVM-mCherry-Cep63 alone generates an ectopic aggregate that can recruit endogenous Cep152, PIK4, and Sas6 (Arrowhead). However, it fails to generate a higher-order cylindrical architecture, likely due to an insufficient amount of Cep152 available in the cytosol to generate the Cep63-Cep152 complex, the building block for the cylindrical assembly. A paired arrows indicate respective signals at the endogenous centrosomal site.

Reviewer #3's comment #6:

PIK4 localization quantification for Fig. 3h-i

PIK4 localization quantification for Fig. 3c-f

Reviewers' Comments:

Reviewer #1:

Remarks to the Author:

The authors have addressed most of my points and – as stated before – this paper is an important contribution to the field and beyond (and I want to see it published). However, I want to see one more experiment that I had originally asked for, if Nature Commun. allows for this. If not, the corresponding point (stability vs high exchange rate) needs to be heavily qualified in the revised manuscript and discussed in a more controversial way than it is done currently.

I am still baffled by the behavior of the microscopic assemblies and (without experimental evidence) not convinced by the explanation offered by the authors in their rebuttal. On the one hand the assemblies are stable for days in buffer, but at the same time have a high exchange rate by FRAP. Thus, the off rate has to be really low, which is of course entirely possible. However, it is thermodynamically and kinetically then hard to see how the subunits from the middle region of the cylinder could ever exchange with the surroundings. Therefore, exchange should preferentially occur at the end, as the authors agree in their rebuttal. I had originally asked the authors to address whether this is indeed the case. Technically, this should be feasible (and fast) to do, e.g. by floating in complexes labelled with a different colour (e.g. blue) and shortly later visualising where the blue labels are on the existing assemblies. The height of the assemblies is such that it would be easy to resolve such a "cap".

Thus, the authors should do this rather than relying on speculation. It is a really important point to address in order to be able to judge what these higher order assemblies are, as there is no associated structural information on them. In the revised manuscript the authors state (concerning the surprising dynamicity of the FRAP experiment) that "the binding energy of incoming molecules could be harnessed to disrupt the preexisting molecular interaction, thus allowing the exchange and recovery of the fluorescence intensity". In order to harness any binding energy, binding would have to occur to begin with. And for binding to occur in the middle of the cylinder, the preexisting subunit would have to leave first. How could they do this, if the assembly is essentially rock-stable? Thus, this is not an explanation for any (significant) potential exchange occurring there. If exchange however occurs preferentially at ends, this could reconcile the baffling FRAP and stability experiments nicely.

My original comment: "It is not clear to me why the assemblies with a larger diameter are (in absolute terms) longer than those with smaller diameter (Figure 2E). Please explain. One would expect their absolute height to be lower, given that way more subunits have to be added per rise."

The authors' rebuttal comment that different stabilities might be at the heart of this is a possibility. However, these assemblies are remarkably stable (for days) and the stability argument looks therefore not really convincing. Do smaller diameter assemblies disappear faster under wash-out / buffer only conditions compared to larger diameter ones? The authors have done the stability experiment now and therefore could provide these data by simple reanalysis. Please comment / discuss shortly in the revised manuscript.

My original comment: "'requires unusual biochemical and biophysical properties". To my mind there is nothing particularly unusual about the interaction that is described in this manuscript. I would remove that sentence."

The authors really should change that and not leave it as they decided to do. None of the described molecular interactions in their high-resolution structure are particularly unusual. The higher order assembly may have surprises, but the authors have no structural information on it to back up that claim. The authors' rebuttal statement that microtubules are one dimensional assemblies is wrong. Microtubule protofilaments could be considered one dimensional, the microtubule itself is three dimensional and helical. Furthermore, I can think of plenty of examples

for proteins that cooperatively self-assemble into a three-dimensional cylindrical architectures, such as Flagellin and others.

Reviewer #3:

Remarks to the Author:

In their revised version the authors have answered some of the questions of this reviewer. The experiments described have been improved, although I think that this paper would greatly benefit, as outlined below, if studies with disease related mutations could be incorporated to show an effect on both the interplay between Cep63 and Cep152 to generate the cylindrical self-assembly and the later consequence of this process on centriole biogenesis.

In Figure 1b, I am still surprised how much Cep63 is precipitated. The authors write on page 5 "the provision of Cep63 or Cep152 greatly (~8-fold) enhanced the homomeric Cep152–Cep152 or Cep63–Cep63 interaction, respectively. However, it seems that the amount of the Cep63-Cep63 precipitated is much more and not stoichiometric to the Cep152-Cep152 homodimer? How was this experiment quantified?

The authors state that there are no MPCH mutations described in either Cep152 or Cep63 that could affect their dimerization or impair PCM assembly. The impact of the studies described in their manuscript would greatly benefit from an analysis of a disease related somatic mutation. In one of their recent publications (Park et al. 2014, Nat Struct. Biol) the authors describe such a mutation in the Cep152 gene affecting centriole assembly. Although this particular mutation is not included in the Md4 region of Cep152, there exist a number of other Cep152 somatic mutations, interestingly also in the Md4 region of Cep152 (see catalog of somatic mutations provided by the Sanger Institute). Does a mutation in this region that has been identified in human cancers affect self-assembly and centriole assembly?

Responses to the reviewers' comments:

Outlined below are our point-by-point answers to the reviewers' comments.

Reviewer #1 (Remarks to the Author):

The authors have addressed most of my points and – as stated before – this paper is an important contribution to the field and beyond (and I want to see it published). However, I want to see one more experiment that I had originally asked for, if Nature Commun. allows for this. If not, the corresponding point (stability vs high exchange rate) needs to be heavily qualified in the revised manuscript and discussed in a more controversial way than it is done currently.

I am still baffled by the behavior of the microscopic assemblies and (without experimental evidence) not convinced by the explanation offered by the authors in their rebuttal. On the one hand the assemblies are stable for days in buffer, but at the same time have a high exchange rate by FRAP. Thus, the off rate has to be really low, which is of course entirely possible. However, it is thermodynamically and kinetically then hard to see how the subunits from the middle region of the cylinder could ever exchange with the surroundings. Therefore, exchange should preferentially occur at the end, as the authors agree in their rebuttal. I had originally asked the authors to address whether this is indeed the case. Technically, this should be feasible (and fast) to do, e.g. by floating in complexes labelled with a different colour (e.g. blue) and shortly later visualising where the blue labels are on the existing assemblies. The height of the assemblies is such that it would be easy to resolve such a “cap”. Thus, the authors should do this rather than relying on speculation. It is a really important point to address in order to be able to judge what these higher order assemblies are, as there is no associated structural information on them.

Answer: As suggested, we generated the Cep63 P1•Cep152 M4d self-assemblies decorated with FITC, washed, and then incubated in the assembly buffer containing the mCherry-Cep63 P1•mGFP-Cep152 M4d complex (see the Supplementary Fig. 2l legend and Methods in page 33 for details). Our results showed that although the mCherry fluorescence from the incoming complex was somewhat enriched at the longitudinal ends of the cylinders, it was not sharply confined to these regions. The results are provided in the **NEW Supplementary Fig. 2l** and described in page 10 line 169. One possibility is that our cylinders are not regularly assembled as observed in actin filaments or microtubules and are built through relatively weak interactions among building block molecules, thus allowing efficient exchanges with incoming molecules (see Supplementary Fig. 2m below). Better understanding the rules of this cylindrical architecture may require substantial amounts of additional works, as noted in the statement from Dr. Alex Mogilner (New York University), who developed the elastic Brownian ratchet mechanism to explain actin-based cell motility.

In the revised manuscript the authors state (concerning the surprising dynamicity of the FRAP experiment) that “the binding energy of incoming molecules could be harnessed to disrupt the preexisting molecular interaction, thus allowing the exchange and recovery of the fluorescence

intensity". In order to harness any binding energy, binding would have to occur to begin with. And for binding to occur in the middle of the cylinder, the preexisting subunit would have to leave first. How could they do this, if the assembly is essentially rock-stable ? Thus, this is not an explanation for any (significant) potential exchange occurring there. If exchange however occurs preferentially at ends, this could reconcile the baffling FRAP and stability experiments nicely.

Answer: As the reviewer suggested, we rephrased the section after consulting with Dr. Alex Mogilner and Dr. José Faraldo-Gómez (NIH), and eliminated the statement that “the binding energy of incoming molecule could be harnessed”. This is shown in page 10 line 164. At present, although we do not know whether the cylinders are rock-stable, at least their disassembly process appears to be kinetically very slow in the assembly buffer. This finding is now confirmed in the **NEW Supplementary Fig. 2m**.

My original comment: “It is not clear to me why the assemblies with a larger diameter are (in absolute terms) longer than those with smaller diameter (Figure 2E). Please explain. One would expect their absolute height to be lower, given that way more subunits have to be added per rise.” The authors’ rebuttal comment that different stabilities might be at the heart of this is a possibility. However, these assemblies are remarkably stable (for days) and the stability argument looks therefore not really convincing. Do smaller diameter assemblies disappear faster under wash-out / buffer only conditions compared to larger diameter ones ? The authors have done the stability experiment now and therefore could provide these data by simple reanalysis. Please comment / discuss shortly in the revised manuscript.

Answer: As recommended, we measured the fluorescence intensities of the mCherry-Cep63 P1•mGFP-Cep152 M4d self-assembly after treating them with buffer or a nonfluorescent Cep63 P1•Cep152 M4d complex for 3 h and quantified them in three different diameter groups (< 300 nm, 300 – 800 nm, and > 800 nm). However, we did not observe significant diameter-dependent stability differences. The results are provided in the **NEW Supplementary Fig. 2m (right)**. At this point, we do not know why the cylinders with a smaller footprint are shorter than the cylinders with a larger footprint. As suggested, our speculations about this counterintuitive observation are provided in page 8 line 119.

Strikingly, however, samples treated with the nonfluorescent Cep63 P1•Cep152 M4d complex exhibited drastically reduced fluorescent intensities for all three groups of cylinders. This finding further supports the FRAP exchange observed in the cylindrical mCherry-Cep63 P1•mGFP-Cep152 M4d self-assembly in Fig. 2j. This finding shown in the **NEW Supplementary Fig. 2m (the 3rd panel, left, and the 3rd column, right)** are described in page 11 line 175 (In a related experiment, we observed----). In the absence of any structural details about how the cylindrical architecture is assembled with the heterotetrameric building block, it is difficult to know how exactly the Cep63 P1•Cep152 M4d complex can kinetically and/or thermodynamically induce the disappearance of the fluorescence signals from the mCherry-Cep63 P1•mGFP-Cep152 M4d self-assemblies. Extensive additional studies and computational modeling works will be necessary to address this important question.

My original comment: ““requires unusual biochemical and biophysical properties”. To my mind there is nothing particularly unusual about the interaction that is described in this manuscript. I

would remove that sentence.” The authors really should change that and not leave it as they decided to do. None of the described molecular interactions in their high-resolution structure are particularly unusual. The higher order assembly may have surprises, but the authors have no structural information on it to back up that claim. The authors’ rebuttal statement that microtubules are one dimensional assemblies is wrong. Microtubule protofilaments could be considered one dimensional, the microtubule itself is three dimensional and helical. Furthermore, I can think of plenty of examples for proteins that cooperatively self-assemble into a three-dimensional cylindrical architectures, such as Flagellin and others.

Answer: As suggested, we took out “unusual” from the text. We meant to state that the ability of a small helical bundle to self-assemble a macroscale cylindrical architecture may require an unusual property. Thank you for the suggestion.

Reviewer #3 (Remarks to the Author):

In their revised version the authors have answered some of the questions of this reviewer. The experiments described have been improved, although I think that this paper would greatly benefit, as outlined below, if studies with disease related mutations could be incorporated to show an effect on both the interplay between Cep63 and Cep152 to generate the cylindrical self-assembly and the later consequence of this process on centriole biogenesis.

In Figure 1b, I am still surprised how much Cep63 is precipitated. The authors write on page 5 “the provision of Cep63 or Cep152 greatly (~8-fold) enhanced the homomeric Cep152–Cep152 55 or Cep63–Cep63 interaction, respectively. However, it seems that the amount of the Cep63-Cep63 precipitated is much more and not stoichiometric to the Cep152-Cep152 homodimer? How was this experiment quantified?

Answer: First of all, we were sorry that we did not precisely understand the reviewer’s comment during our first revision. We revisited our experimental scheme for Fig. 1b and Supplementary Fig. 1j, and found that these two experiments were carried out using a different transfection format. Because it is very difficult to express enough amounts of the full-length Cep63 and Cep152 forms (especially the high MW Cep152) even by transfection, HEK293 cells were transfected individually (so that we can use more DNA). Cell lysates prepared from each transfection were mixed and then subjected to immunoprecipitation analysis. As a result of the procedural differences between these experiments and others in the manuscript, the gels shown in Fig. 1b and Supplementary Fig. 1j exhibit a very low level of homomerization between FLAG-Cep152 and HA-Cep152 (Fig. 1) or FLAG-Cep63 and HA-Cep63 (Supplementary Fig. 1j). This is because individually expressed Cep63 and Cep152 could have been already homomerized as shown in Supplementary Fig. 1g and 1h. On the other hand, since the heteromerization activity between Cep63 and Cep152 are undiminished under these experimental formats, we can observe much stronger HA-Cep63 signal for FLAG-Cep152 IP (Fig. 1b) and HA-Cep152 signal for FLAG-Cep63 IP (Supplementary Fig. 1j) than their respective homomeric interactions. To avoid misleading conclusions, we now provided detailed experimental procedures and offered our explanations in the Fig. 1b and

Supplementary Fig. 1j legends. Thank you for recognizing the problem in our initial manuscript.

In an independent experiment, we assessed the efficiency of the Cep63 FL-Cep152 FL interaction by carrying out coimmunoprecipitation followed by silver staining (**NEW Supplementary Fig. 1i**). As we see in the gel, FLAG-Cep63 efficiently coprecipitated HA-Cep152. This is stated in page 5 line 54. However, because of the homomerization capacity of both Cep63 and Cep152 and their unknown levels of soluble forms in the lysates (both Cep63 and Cep152 are highly aggregative!), the significance of this result in predicting the stoichiometry of the Cep63-Cep152 interaction is very limited. This point is stated in the figure legend.

The authors state that there are no MPCH mutations described in either Cep152 or Cep63 that could affect their dimerization or impair PCM assembly. The impact of the studies described in their manuscript would greatly benefit from an analysis of a disease related somatic mutation. In one of their recent publications (Park et al. 2014, Nat Struct. Biol) the authors describe such a mutation in the Cep152 gene affecting centriole assembly. Although this particular mutation is not included in the Md4 region of Cep152, there exist a number of other Cep152 somatic mutations, interestingly also in the Md4 region of Cep152 (see catalog of somatic mutations provided by the Sanger Institute). Does a mutation in this region that has been identified in human cancers affect self-assembly and centriole assembly?

Answer: We agree with the reviewer that the effect of cancer-associated mutation(s) on the assembly of the Cep63-Cep152 complex will be very important. However, we believe a lot of other studies should be proceeded before we get into this issue. First, we need to establish more reliable assays to examine the kinetics and dynamics of assembly/disassembly processes for our cylinders. Second, we need to know whether the mutations reported by the Sanger Institute are validated disease-causing mutations (not just mutations found in cancer patients). While the research direction that this reviewer offers is a very valuable one, we feel that the proposed experiment is beyond the scope of this manuscript.

Sincerely,

Kyung Lee

Reviewers' Comments:

Reviewer #1:

Remarks to the Author:

The authors have addressed my points. The exact nature of the observed assemblies remains unclear and some of their properties are rather unusual. However, this is now experimentally addressed and clearly discussed in the manuscript. I have no doubt that this paper will stimulate the field and recommend publication.

One small thing to fix is the following sentence (Page 8)

"It is possible that the self-assemblies with smaller diameters (i.e., smaller bases for cylindrical structures) are either physically unstable or structurally unfavored during the assembly process. "

The corresponding figure (Supplementary Figure 2m) is not referred to. Please add a sentence referring to this figure, eg:

"Their disassembly kinetics, however, appear unchanged compared to the larger diameter assemblies (Supplementary Figure 2m).

Reviewer #3:

Remarks to the Author:

The authors have partially answered this reviewer's comments on Cep63 and Cep152 interaction shown now in Figure S1i of their revised manuscript. Although I admit that this manuscript contains interesting data on cooperative selfassembly of a Cep63 and Cep52 dependent new structural entity, I still think that a broader readership would greatly benefit from the analysis of cancer-related mutations impairing this assembly mechanism. I am therefore not convinced and still not in favour of publishing the article in Nature Communications.

Responses to the reviewers' comments:

Outlined below are our point-by-point answers to the reviewers' comments.

REVIEWERS' COMMENTS:

Reviewer #1 (Remarks to the Author):

The authors have addressed my points. The exact nature of the observed assemblies remains unclear and some of their properties are rather unusual. However, this is now experimentally addressed and clearly discussed in the manuscript. I have no doubt that this paper will stimulate the field and recommend publication.

One small thing to fix is the following sentence (Page 8)

"It is possible that the self-assemblies with smaller diameters (i.e., smaller bases for cylindrical structures) are either physically unstable or structurally unfavored during the assembly process. "

The corresponding figure (Supplementary Figure 2m) is not referred to. Please add a sentence referring to this figure, eg:

"Their disassembly kinetics, however, appear unchanged compared to the larger diameter assemblies (Supplementary Figure 2m).

Answer: We agree with the reviewer's assessment that the disassembly kinetics for the self-assemblies could be similar based on the data shown in Figure S2m and are likely independent of the size of their cylindrical diameters. However, we also concern that it could be an oversimplified statement at present. In our view, more thorough analyses with time-lapse intensity measurements under multiple disassembling schemes would be necessary to yield any solid conclusion. Therefore, instead of making any premature conclusion, we think it would be the best to leave the manuscript as it is. Thank you for the suggestion.

Reviewer #3 (Remarks to the Author):

The authors have partially answered this reviewer's comments on Cep63 and Cep152 interaction shown now in Figure S1i of their revised manuscript. Although I admit that this manuscript contains interesting data on cooperative selfassembly of a Cep63 and Cep52 dependent new structural entity, I still think that a broader readership would greatly benefit from the analysis of cancer-related mutations impairing this assembly mechanism. I am therefore not convinced and still not in favour of publishing the article in Nature Communications.

Answer: We agree with the reviewer that investigating the effect of cancer-associated mutation(s) on the kinetics and dynamics of the Cep63-Cep152 self-assembly will be very important. However, as we indicated in our previous rebuttal letter, we may need a better

time-controlled, robust assay platform to address this important question. We believe that the proposed experiment is beyond the scope of this manuscript.

Sincerely,